# CSE-ET: Joint Optimization for 3D Multi-UAV Explore-and-Track under Connectivity, Separation, and Energy Constraints

## Abstract

We study cooperative target tracking by UAV swarms in open airspace under realistic resource and safety constraints (limited communication, limited power, and inter-UAV safety separation), and introduce active exploration based on spatial information entropy. Most existing methods focus on one or two subtasks (e.g., only tracking accuracy, or treating exploration/communication/power as independent modules), and their metrics and scenario coverage are limited, making it difficult to reflect practical performance under the above constraints. Targeting an evading UAV with an optimal avoidance policy, this paper jointly considers communication, spatial exploration, power consumption, and inter-UAV collision avoidance for the tracking team, and designs a unified joint-optimization framework based on multi-agent reinforcement learning. The framework models low-level action control with a base model, high-level behavior control with a multi-agent decision model, and is updated by a joint optimization algorithm. In simulation, we construct evaluation metrics covering tracking, power, and exploration, and compare with the mainstream multi-agent baselines. Experiments show that our method outperforms the baselines on average by **+23%** in tracking success rate, **+25%** in power saving, **+24%** in spatial exploration, and **+18%** in aggregate reward. To our knowledge, this paper presents the 3D multi-UAV cooperative tracking framework that is most closely aligned with practical constraints.

## 1 Introduction

Multi-UAV Target Tracking (Akin, 2023; Zhang et al., 2023b; Dong et al., 2019) has attracted significant research attention due to its crucial applications in public safety and military missions. This technology leverages collaborative target perception, swarm communication, and coordinated tracking by multiple unmanned aerial vehicles (UAVs) to achieve *persistent target tracking* (Zhang et al., 2023a).

Existing approaches predominantly rely on information fusion to predict target trajectories (Pi et al., 2016). These methods estimate the target state distribution via multi-sensor data. UAVs can be equipped with RF sensors to measure signal characteristics, integrating these measurements with filtering techniques for target localization and tracking (Sung & Tokekar, 2022). However, this approach depends on the assumption of a stationary distribution for the target state. In practical scenarios, particularly those involving adverse radio frequency environments (Shih et al., 2018), intermittent connectivity frequently occurs between the target and UAVs. While sampling-based methods (Koohifar et al., 2018) can partially mitigate this issue, their stochastic sampling nature is prone to causing trajectory jitter.

Solutions combining deep learning techniques for image processing (Bouguettaya et al., 2022; Yao et al., 2025) with UAV control have been proposed by introducing the cascaded framework of target re-identification and association tracking, which suffers from significant error accumulation and latency, as well as inadequate swarm coordination capabilities. Inspired by the application of Multi-Agent Reinforcement Learning (MARL) to agent cooperative control (Jia et al., 2025; Wang et al., 2025) and the successful development of next-generation communication networks (Han et al., 2016; Zhang et al., 2024), we propose a Joint Optimization Framework for complex coordinated tracking

by multi-agents in pursuit-evasion games. Our design addresses the above limitations through three integrated levels from fundamental to advanced: a Base Model, a Decision Model, and a Joint Optimization Algorithm. Our contributions can be summarized as follows:

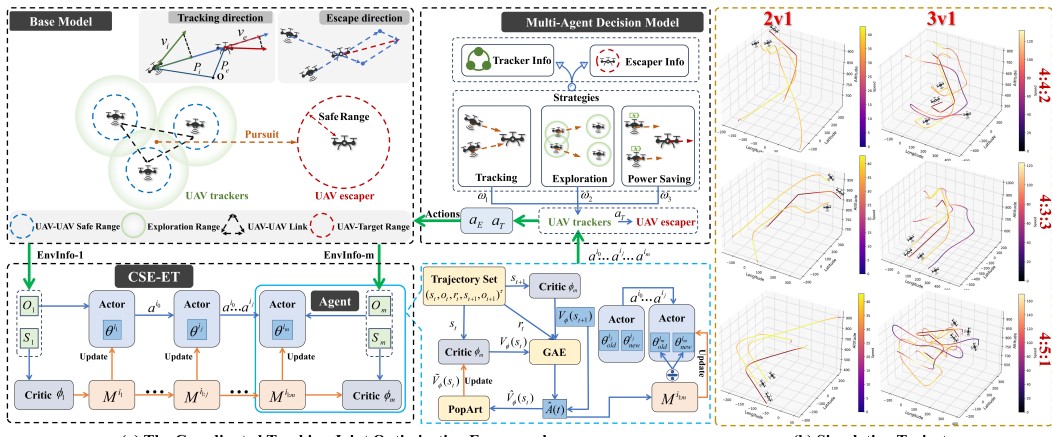

(a) The Coordinated Tracking Joint Optimization Framework          (b) Simulation Trajectory

Figure 1: (a): The Framework bridges the interaction between the pursuit-evasion environment **(Base Model)** and the coordinated policies **(Decision Model)**, enabling holistic update iterations through **joint optimization algorithms (CSE-ET)**. (b): Different reward ratios (Section 3.3) make agents have different tracking characteristics. The speed of the agent (4:4:2) is closer to the target. Agents (4:3:3) have the characteristics of outward expansion. The tendency of the agent (4:5:1) is tracked directly. See Figure 6 for the simulation tracks of other algorithms.

- For a partially observable multi-agent decision problem under limited communication, limited power, and safety-separation constraints, we propose a constraint cooperative target-tracking learning framework for open airspace that integrates spatial information–entropy–driven active exploration and power efficiency into the tracking process.

- For the complex behavior trade-off between exploration and power in multi-UAV cooperative tracking, we construct a tracking–exploration–power multi-objective reward and a curriculum-style fusion strategy, which significantly improves training stability and sample efficiency, and we provide systematic ablations.

- To address the learning stability of coordinated behaviors under multi-objective rewards in multi-agent systems, we propose a multi-agent joint optimization algorithm. Each agent adopts an independent recurrent neural policy network combined with an advantage decomposition lemma and sequential policy parameter updates, which—compared with shared-update MAPPO—effectively prevents training collapse caused by inter-module policy conflicts. We conduct a comprehensive evaluation against multi-agent baselines MAPPO(Yu et al., 2021), HAPPO, and HATRPO(Kuba et al., 2022). Experiments show that our method outperforms the baselines on average by +23% in tracking success, +25% in power savings, +24% in spatial exploration, and +18% in aggregate reward. The released framework code includes agent training and evaluation to facilitate reproducibility and comparison.

## 2 BASE MODEL

The Base Model includes tracking, escape, communication and power consumption models, and the most basic agent interaction logic is modeled.

### 2.1 UAV TRACKING MODEL

The single UAV adopts the 3D dynamic model:

$$V_x(t) = V_x(t-1) + a(t-1)\cos\phi_b\sin\varphi_b\Delta t, \tag{1a}$$

$$V_y(t) = V_y(t-1) + a(t-1)\cos\phi_b\cos\varphi_b\Delta t, \tag{1b}$$

$$V_z(t) = V_z(t-1) + a(t-1)\sin\phi_b\Delta t. \tag{1c}$$

In this context, $V(t)$, $P(t) = (x(t), y(t), z(t))$, and $a(t)$ represent the velocity vector, position vector and acceleration vector. The angle $\phi_b(t)$ denotes the body angle between the velocity vector and the $o\text{-}x\text{-}y$ plane. The angle $\varphi_b(t)$ represents the angle between the projection of the velocity vector $v'$ onto the $o\text{-}x\text{-}y$ plane and the $o\text{-}y$ axis.

## 2.2 UAV ESCAPE MODEL

The escape model is flexibly adjusted based on the position $P_i(t)$ of the tracking UAVs, as shown in Figure 1.a (Escape direction):

$$P_e(t) = P_e(t-1) + \sum_{i=1}^{N} \frac{D_\alpha - P_i(t-1)}{\|D_\alpha - P_i(t-1)\|_2} \cdot \Delta t, D_\alpha = \sum_{l=0}^{L} P_e(t-l) \cdot \alpha^l, \tag{2}$$

where $P_e(t)$ is the current position of the escape UAV, $p$ is the distance amplification parameter and $\alpha$ is the discount factor. $D_\alpha$ modulates evasion difficulty via sliding-window smoothing over the evader's position history (window length $L$). Smaller $L$ yields a more reactive evader, whereas larger $L$ dampens agility. Results are reported in Fig.2a.

## 2.3 U2U COMMUNICATION MODEL

The U2U communication model in (Zhang et al., 2019) is adopted and the UAV group is described by the time-varying graph. The adjacency matrix of a graph is denoted as $A_G(t) = a_{ij}(t)$. Using $A_G(t)$, the graph Laplacian matrix is defined as follows:

$$[L_A]_{ij}(t) = \begin{cases} -\mathrm{SNR}_{ij}(t), & i \neq j, \\ \sum_{k=1,k\neq i}^{N} \mathrm{SNR}_{ik}(t), & i = j, \end{cases} \tag{3}$$

A positive second smallest eigenvalue $\lambda_2(t) > 0$ implies the connectivity of the graph. Let $a_{ij}(t)$ denote the connectivity quality between UAVs $i$ and $j$. The mapping relationship between connectivity quality and distance is characterized via the Signal-to-Noise Ratio (SNR) as follows:

$$\mathrm{SNR}_{ij} = \frac{GP}{\sigma^2} d_{ij}^{-\delta} \mathbf{1}\{d_{ij} \leq \beta\}, \quad \beta = \left(\frac{GP}{\sigma^2(2^{\zeta_{\mathrm{th}}} - 1)}\right)^{1/\delta}, \tag{4}$$

the physical is considered connected when $\mathrm{SNR}_{ij}(t) \geq \gamma_{\mathrm{th}}$ (equivalently $d_{ij}(t) \leq \beta$). The threshold $\beta$ can be derived from Shannon inequality. Where $d_{ij}$ is the distance between the inter-UAV distance, $G$ is the power gain caused by amplifiers and antennas, $\delta$ is the channel fading factor, and $P$ is the same signal transmission power, $\sigma^2$ is the variance of the Gaussian white noise, $\zeta_{\mathrm{th}}$ is the communication threshold.

## 2.4 UAV POWER CONSUMPTION MODEL

UAV power consumption comprises propulsion generation and communication (the latter being negligible according to (Yang et al., 2021a)). Following (Zeng et al., 2019), the average propulsion power consumption for rotorcraft UAVs in three-dimensional scenarios can be expressed as:

$$\bar{P}_c(t) = \frac{1}{N} \sum_{k=1}^{N} P_c(v(t)), \tag{5}$$

where $P_c(v) = \widehat{P}_0(v) + \widehat{P}_i(v) + \widehat{P}_b(v)$. The power consumption of UAV $N$ at time $t$ is expressed:

$$\widehat{P}_0(v) = P_0(1 + \frac{3v^2}{U_{tip}^2}), \widehat{P}_i(v) = P_i(\sqrt{1 + \frac{v^4}{4v_0^4}} - \frac{v^2}{2v_0^2})^{1/2}, \widehat{P}_b(v) = \frac{1}{2}\rho\tau\varphi Av^3, \tag{6}$$

where $\widehat{P}_o(v)$, $\widehat{P}_i(v)$, and $\widehat{P}_b(v)$ respectively denote the blade profile, induced power, and parasitic power; $P_0$ and $P_i$ are two constants representing blade profile power and induced power in hovering state. $U_{\text{tip}}$ represents the tip speed of the rotor blade, $v_0$ is the average induced velocity in the hover state, $\varphi$ represents the fuselage drag ratio, $\tau$ represents the rotor solidity; $\rho$ is the air density; and $A$ represents the rotor disc area.

This paper adopts the closed-form propulsion power model for rotary-wing UAVs given in [Zeng]. The model is simplified from the more general $P(V, \kappa)$ under the assumptions of straight, near-level flight and a thrust-to-weight ratio close to 1. In this work, the UAV executes smooth 3D trajectories under velocity and acceleration constraints in the action space with a time step of $\Delta t = 0.01s$. The experiments in Fig. 7 indicate that Eq. 6 is a reasonable engineering approximation for characterizing instantaneous propulsion power. For extreme maneuvers such as aggressive climb/dive, incorporating the full $\kappa$-dependent model or additional potential-energy terms is a meaningful extension, but beyond the scope of this paper.

# 3 MULTI-AGENT DECISION MODEL

The decision model incorporates state space, and a comprehensive reward function derived from the foundational models, which abstracts states from the foundational layer to model desired behaviors.

## 3.1 STATE SPACE

The state space characterizes the operational status of UAV tracking. The quantities of pursuing and evading UAVs are denoted as $N$ and $E$, respectively.

$$S_k(t) = (v_k(t), P_k(t), \alpha_U, \alpha_T), \forall k \in (N + E). \tag{7}$$

We leverage the relative relations among elements in $S_k$ to increase attention to key information, as shown in Table 1.

Table 1: The state space of an individual UAV. Where $\alpha_U$ is the angle between the vector $v_i(t)$ and the vector $P_e(t) - P_i(t)$, $\alpha_T$ is the angle between the vector $v_e(t)$ and the vector $P_e(t) - P_i(t)$. $v_{min}$ and $v_{max}$ respectively denote the maximum and minimum velocity, $d_{\max}^{ut}$ is the maximum distance from the initial position.

| Observation | Dimension |
|:---:|:---:|
| $(v_e(t) - v_i(t))/(v_{\max} - v_{\min})$ | 3 |
| $(P_e(t) - P_i(t))/d_{\max}^{ut}$ | 3 |
| $\alpha_U/2\pi$ | 1 |
| $\alpha_T/2\pi$ | 1 |

## 3.2 ACTION SPACE

Compared to fixed-wing UAVs, rotorcraft UAVs achieve omnidirectional movement without altering aircraft attitude (Urakubo et al., 2023) and maintain stationary hovering. Consequently, the action space is defined by three-dimensional acceleration vectors, simplifying yaw angle computation.

$$Action = [a_x, a_y, a_z]^N. \tag{8}$$

## 3.3 REWARD FUNCTION

The speed $\eta_A$, position $\eta_B$, exploration $\eta_C$ and power consumption $\eta_D$ of the pursuit-evasion UAVs are considered as:

$$R = \omega_1(\eta_A + \eta_B) + \omega_2\eta_C + \omega_3\eta_D, \tag{9}$$

where $\omega_1, \omega_2, \omega_3$ is an adjustable parameter and satisfies $\omega_1 + \omega_2 + \omega_3 = 1$.

### 3.3.1 TRACKING REWARD

First, the tracking advantage is defined when the projection of velocity vector $v_i$ onto $P_e(t) - P_i(t)$ exceeds the projection of $v_e$ onto $P_e(t) - P_i(t)$, while also accounting for the speeds and angles of the pursuing and evading UAVs, as shown in Figure 1.a (Tracking direction):

$$\eta_A = \frac{1}{N} \sum_{i=1}^{N} \frac{v_i \left( P_e(t) - P_i(t) \right)}{\| P_e(t) - P_i(t) \|} - \frac{v_e \left( P_i(t) - P_e(t) \right)}{\| P_i(t) - P_e(t) \|}. \tag{10}$$

Second, the distance reward $\eta_B$ is defined as follows, where $d_{\min}^{ue}$ denotes the minimum safe distance of the pursuing and evading UAVs:

$$\eta_B = [\, 1 + \frac{1}{N} \sum_{i=1}^{N} \left( \| P_e(t) - P_i(t) \| - d_{\min}^{ue} \right)^2 \,]^{-1}. \tag{11}$$

### 3.3.2 SPACE EXPLORATION RAWARD

Under guaranteed communication quality, the closer the UAVs are, the greater the overlap in their sensed information. Therefore, we extend the spatial information-entropy metric (Batty, 1974) from 2D to 3D and use it as the primary component of the spatial exploration reward. Within the tracking UAV swarm, assume that UAV $i$ relative to UAV $j$ at time $t$ is represented by polar coordinates $(r_t, \theta_t, \varphi_t)$. The information-correlation density function is defined as follows:

$$\zeta \left( r, \theta, \varphi, l \right) = K(l) \left( e^{-\lambda r} + \frac{1}{\lambda} \delta_{r,\theta,\varphi} \left( r_t, \theta_t, \varphi_t \right) \right), \tag{12}$$

where $\lambda$ is the strength parameter, $l$ is a variable parameter indicating the maximum inspection distance. The intuition behind this definition of the information-correlation density is twofold. First, $e^{-\lambda r}$ indicates that when UAVs are close, their acquired information highly overlaps, and this correlation quickly weakens as their distance increases. Second, because the target's information is critical, we assign a special point weight at the target—implemented by $\delta_{r,\theta,\varphi} \left( \cdot \right)$ (the Dirac delta function)—to highlight its importance.

$K(l)$ is the normalized function of $l$. When $0 \le r_t \le l$, the $\delta$ term contributes a constant 1 to the normalization (independent of $(r_t, \theta_t, \varphi_t)$), and thus $\bar{K}$ is only a function of $l$. *See Appendix A.2.1 for detailed derivation.*

$$K = \frac{\lambda}{4\pi\lambda\kappa(l) + 1}, \quad \kappa(l) := \int_0^l r^2 e^{-\lambda r} dr. \tag{13}$$

Similar to information entropy, the spatial information entropy corresponding to information correlation density is defined as:

$$\mathcal{H}(l) = - \int_0^\pi \int_0^l \int_0^{2\pi} r^2 \zeta \left( r, \theta, \varphi, l \right) \ln \zeta \left( r, \theta, \varphi, l \right) dr d\theta d\varphi. \tag{14}$$

Obtained by algebraic calculation, *see Appendix A.2.2 for detailed derivation.*

$$\mathcal{H}(l) = 4\pi K(l) \left[ \left( 3 - \ln K(l) \right) \kappa(l) - l^3 e^{-\lambda l} \right] + K(l) h \left( \ln K(l) - \lambda r_t \right). \tag{15}$$

Since $\kappa'(l) = l^2 e^{-\lambda l} > 0$ and $\kappa(l)$ increases monotonically with $l$, $K(l)$ decreases monotonically with $l$ and converges to a finite limit. Finally, we obtain that $\mathcal{H}(l)$ increases monotonically with $l$ and then saturates, which accords with the intuition that, during exploration, information acquired by different UAVs evolves from overlapping to fully independent. We now formalize the spatial exploration reward design, subject to the simultaneous satisfaction of the minimum inter-UAV safety distance $d_{\min}^{uu}$ for collision avoidance and the connectivity condition in (3):

$$\lambda_2(t) > 0, \quad d_{ij} \ge d_{\min}^{uu}, \forall i, j \in N, i \ne j. \tag{16}$$

The reward function can be expressed as follows:

$$\eta_C = \frac{1}{1 + \left( \bar{d}_N(t) - \mathcal{H}(l) \right)^2}, \quad \bar{d}_N(t) = \frac{2}{N(N-1)} \sum_{i \ne j} d_{ij}(t), \tag{17}$$

where $\bar{d}_N(t)$ represents the average distance between UAVs.

### 3.3.3 POWER CONSUMPTION REWARD

Current rotorcraft UAVs primarily rely on battery power. As specified in (6), the swarm should maintain velocity proximity to the evading UAV upon entering the designated tracking zone. First, within the maximum tracking range $\rho_s$, a minimum safety distance $d_{\min}^{ut}$ should be maintained between the pursuing and evading UAVs to prevent collisions.

$$d_{ie} \geq d_{\min}^{ue}, d_{ie} \leq \rho_s, \quad \forall i \in N. \tag{18}$$

Second, we need to ensure connectivity in (3). The power consumption reward can be expressed as follows:

$$\eta_D = [1 + (\bar{P}_c(t) - P_c(v_e))^2]^{-1}. \tag{19}$$

### 3.4 TWO-STAGE CURRICULUM LEARNING

We design a two-phase tracking:

1. Accelerated tracking implements tracking rewards defined in (10) and (11).
2. Stabilized tracking emphasizes exploratory and power-conserving behaviors, activated when conditions in (16), (18) are met. This phase augments the reward structure with (17) and (19).

This is achieved by resetting the environment for agents exceeding boundary conditions while imposing negative rewards via (19):

$$\begin{cases} R - 40, \exists d_{it} > 350 \, \mathrm{m}, \\ R - 10, \exists d_{it} < 25 \, \mathrm{m}. \end{cases} \tag{20}$$

Consecutive Steps (CS) are configured to calculate the continuous timesteps during which the agent satisfies conditions (16) and (18). In Phase 1, if the environment step count exceeds the Speed up Episode Steps threshold while the Consecutive Length (CL) remains zero — indicating failure to enter Phase 2 — a negative reward is applied and the environment is reset:

$$R - 10, (CS > 400 \cap CL = 0), \tag{21}$$

Consecutive Failed Steps (CFS) are configured to tally continuous timesteps during which the agent fails to satisfy any of conditions (16) and (18). Upon the initial occurrence of $CL \neq 0$, a reward is applied to signify entry into Phase 2. Should CFS exceed 50, a penalty is applied and the environment is reset. Configuring a CFS threshold enables the agent to learn a more responsive tracking strategy.

$$\begin{cases} R + 10, (\text{previous\_CS} \leq 0 \cap \text{CS} > 0), \\ R - 10, (\text{previous\_CS} > 0 \cap \text{CS} == 0 \cap \text{CFS} \geq 50), \end{cases} \tag{22}$$

Reward constraints for exploration and power saving are formulated as follows:

$$\left| \bar{d}_N(t) - \mathcal{H}_{SI}(l) \right| < 20, \tag{23a}$$

$$\left| \bar{P}_c(t) - P_c(v_t) \right| < 5. \tag{23b}$$

A final reward is granted when the cumulative proportion of timesteps satisfying conditions (16), (18) and (23) exceeds 1/3, indicating that all module requirements have been met: $R + 10$.

## 4 MULTI-AGENT COORDINATED TRACKING JOINT OPTIMIZATION

As shown in Figure 1.a (CSE-ET), Value $V_\phi$ is a mapping that critic networks need to learn: $S \to R$. Strategy $\pi_\theta$ is actor network learning a mapping from sampling action A from $N(\mu, \sigma^2)$. In CSE-ET, each UAV owns a dedicated policy network, while the swarm shares a single value network. Using the multi-agent advantage decomposition lemma and a sequential policy update scheme, we avoid policy conflicts among agents. For any ordered agent subset $i_{1:m}$, the state-action value function Q can be defined as:

$$Q_\pi^{i_{1:m}}\left(s, \mathbf{a}^{i_{1:m}}\right) \triangleq \mathbb{E}_{\mathbf{a}^{-i_{1:m}} \sim \pi^{-i_{1:m}}} \left[ Q\left(s, \mathbf{a}^{i_{1:m}}, \mathbf{a}^{-i_{1:m}}\right) \right], \tag{24}$$

the Q value represents the average return of agents $i_{1:m}$ in state $S$ and action group $\mathbf{a}^{i_{1:m}}$. On this basis, the definition of advantage function A is obtained:

$$A_\pi^{i_{1:m}}\left(s, \mathbf{a}^{j_{1:k}}, \mathbf{a}^{i_{1:m}}\right) \triangleq Q_\pi^{j_{1:k},i_{1:m}}\left(s, \mathbf{a}^{j_{1:k}}, \mathbf{a}^{i_{1:m}}\right) - Q_\pi^{j_{1:k}}\left(s, \mathbf{a}^{j_{1:k}}\right). \tag{25}$$

The joint advantage function is decomposed into the sum of local advantages of each agent in the process of sequential update by the multi-agent advantage decomposition lemma:

$$A_\pi^{i_{1:m}}\left(s, \mathbf{a}^{i_{1:m}}\right) = \sum_{j=1}^{m} A_\pi^{i_j}\left(s, \mathbf{a}^{i_{1:j-1}}, a^{i_j}\right), \tag{26}$$

introducing importance sampling to simplify calculation(Kuba et al., 2022):

$$\mathbb{E}_{s \sim \rho^\pi, \mathbf{a} \sim \boldsymbol{\pi}}\left[\frac{\hat{\pi}^{i_m}\left(a^{i_m} \mid s\right)}{\pi^{i_m}\left(a^{i_m} \mid s\right)} \cdot \frac{\bar{\boldsymbol{\pi}}^{i_{1:m-1}}\left(\mathbf{a}^{i_{1:m-1}} \mid s\right)}{\boldsymbol{\pi}^{i_{1:m-1}}\left(\mathbf{a}^{i_{1:m-1}} \mid s\right)} A_\pi(s, \mathbf{a})\right], \tag{27a}$$

$$M_\pi(s, \mathbf{a}) = \frac{\bar{\boldsymbol{\pi}}^{i_{1:m-1}}\left(\mathbf{a}^{i_{1:m-1}} \mid s\right)}{\boldsymbol{\pi}^{i_{1:m-1}}\left(\mathbf{a}^{i_{1:m-1}} \mid s\right)} A_\pi(s, \mathbf{a}), \tag{27b}$$

where $A_\pi(s, \mathbf{a})$ is an advantage function (Schulman et al., 2017).Then the generalized advantage estimator GAE (Schulman et al., 2015) is used to get $A_\pi$. It can be further normalized using PopArt (Hessel et al., 2019) to get $\tilde{V}_\phi(s_t)$. *See Appendix A.5 for detailed steps.* Finally, each agent's strategy can be trained to maximize:

$$L(\theta^{i_m}) = \frac{1}{BT}\sum_{b=1}^{B}\sum_{l=0}^{T}\min\left[\frac{\pi_{\theta^{i_m}}^{i_m}\left(a_t^{i_m} \mid o_t^{i_m}\right)}{\pi_{\theta_k^{i_m}}^{i_m}\left(a_t^{i_m} \mid o_t^{i_m}\right)}, \text{clip}\left(\frac{\pi_{\theta^{i_m}}^{i_m}\left(a_t^{i_m} \mid o_t^{i_m}\right)}{\pi_{\theta_k^{i_m}}^{i_m}\left(a_t^{i_m} \mid o_t^{i_m}\right)}, 1 \pm \epsilon\right)\right]M^{i_{1:m}}(s_l, \mathbf{a}_l). \tag{28}$$

The V-value network can be trained by minimizing:

$$L(\phi) = \frac{1}{BT}\sum_{b=1}^{B}\sum_{t=0}^{T}\left(V_\phi(s_t) - \tilde{V}_\phi(s_t)\right)^2. \tag{29}$$

Through all the above settings, CSE-ET can be summarized as **Algorithm 1**. The process of parameter updating is divided two stages: environment interactive updating stage and neural network updating stage.

## 5 SIMULATION

In this section, we present simulation results and performance analysis of the proposed CSE-ET algorithm across three metrics: tracking success rate, space exploration rate and power saving rate. Experimental findings demonstrate that comparative evaluations against established reinforcement learning baselines reveal CSE-ET's superiority in both reward accumulation and convergence speed. Crucially, adjusting the weighting ratios among sub-models within the reward function enables environment-specific adaptability.

### 5.1 SIMUALTION SETUP

It is assumed that the information of any UAV can be obtained through sensors. The scenario is set in an unbounded 3D domain where the evading UAV is initially positioned at the origin, while pursuing UAVs are randomly distributed within a region defined by the intersection of two circular areas of radii 200 $m$ and 250 $m$ in the xy-plane, with relative z-axis positions constrained within ±150 $m$ of the target. Pursuing UAVs initiate with velocity $V_i$ in arbitrary directions below the maximum velocity threshold, adopting a transmission power of 10 dBm as specified in (Yang et al., 2021b). Safety distances are configured at 25 $m$ for UAV-UAV collision avoidance and 50 $m$ for UAV-target avoidance, where the larger UAV-target separation accommodates enhanced position sharing among adjacent agents and provides critical buffer space against unpredictable target trajectories. Crucially, the evading UAV's acceleration vector is derived by normalizing the distance vector from pursuing UAVs to the target. Agent-environment interactions occur at $\Delta t$ intervals with rewards computed per cycle, while the expected $\mathcal{H}(l)$ equivalent distance in (17) is numerically evaluated as $\hat{d}[\hat{H}]$ to

simulate environmental conditions; Detailed configurations for equations (2), (4), (6), and (17) are provided in Table 4.

We instantiate the simulator in NVIDIA Isaac Gym(Makoviychuk et al., 2021). With a carefully shaped reward and GPU-accelerated physics, a full training run completes in 10 hours on a single RTX 3090. We launch 2,048 parallel environments to maximize throughput while keeping compute usage modest. Detailed parameter settings are provided in Table 5.

## 5.2 PERFORMANCE ANALYSIS

We show the performance comparison of the proposed CSE-ET algorithm with three popular RL baselines: MAPPO, HAPPO and HATRPO. We use the same physical environment to set *Appendix A.3* in the training process. In these performance evaluation results, the $\omega_1 : \omega_2 : \omega_3$ were configured at a 4:3:3 ratio to balance the relative importance of power conservation and exploration. These metrics quantify performance as the ratio of compliant timesteps to Phase 2 submodules timesteps: (1) **Tracking Success Rate** (TSR): Satisfy the conditions (16) and (18). (2) **Exploration Rate** (ER): Satisfy the conditions (16), (18) and (23a). (3) **Power Saving Rate** (PSR): Satisfy the conditions (16), (18) and (23b).

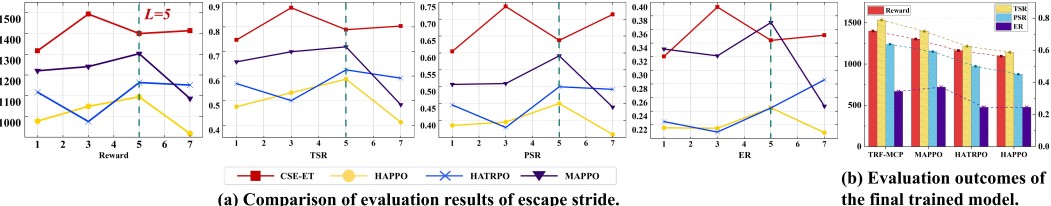

(a) Comparison of evaluation results of escape stride.

(b) Evaluation outcomes of the final trained model.

Figure 2: (a) Assessing tracking algorithm robustness by modulating the evader model's responsiveness.(Larger values of L correspond to diminished evader responsiveness. L=7 loses tracking significance.) **CSE-ET is superior to other models only when the single ER index of L=5 falls behind MAPPO.** (b) At L=5, **CSE-ET is superior to HAPPO(28%, 34%, 41%, 40%), HATRPO(20%, 26%, 27%, 41%) and MAPPO(7%, 10%, 8%, -8%).**

As shown in Figure 2, CSE-ET is superior to other models by modifying $L$ to change the maneuverability of escape model, and is similar to MAPPO in ER. While CSE-ET exhibits optimal performance at $L = 3$, other algorithms achieve peak comprehensive performance at $L = 5$. To validate the robustness of our proposed algorithm, subsequent experiments were conducted under the $L = 5$ configuration.

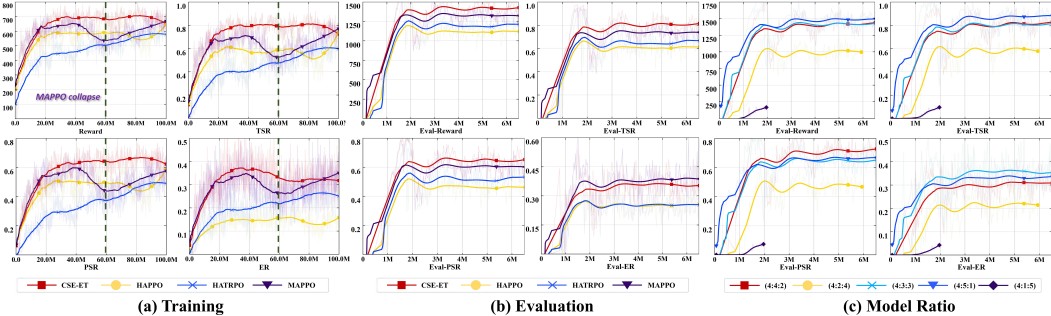

(a) Training

(b) Evaluation

(c) Model Ratio

Figure 3: (a) CSE-ET demonstrates greater stability compared to MAPPO, achieves higher average rewards than HAPPO, and converges faster relative to HATRPO. The x-axis represents the environmental interaction timesteps. (b) Evaluation results (c) Evaluation of five parameter ratios

As illustrated in Figure 3.a, CSE-ET demonstrably outperforms the three baseline methods across all metrics during training. The CSE-ET algorithm stabilizes at 30 million timesteps, maintaining approximate values of 700 for reward, 0.8 for TSR, 0.33 for ER, and 0.65 for PSR. At 60 million

steps, the shared-update MAPPO run temporarily collapsed due to conflicts among submodules in the multi-component reward, and later converged. In sequential updates, the current agent conditions on the previous agent's action as a prior and selects the action that maximizes the global return, avoiding the situation where parallel updates benefit individuals but harm the team overall. The training process alone cannot fully demonstrate algorithmic performance. The following issues require further investigation through model evaluation experiments: (1) At training completion, CSE-ET attains comparable levels to HAPPO and MAPPO in Reward, TSR, and PSR. (2) Whether the decline in CSE-ET's ER adversely impacts overall efficacy.

As shown in Figure 3.b, CSE-ET maintains optimal performance in Reward, TSR, and PSR, while achieving second-best performance in ER. This pattern may be attributed to the algorithmic advantage of shared parameters for spatial exploration over sequential updating approaches.The final result is shown in Figure 2.b.

### 5.3 MODEL ANALYSIS

As shown in Figure 3.c, the (4:1:5) and (4:2:4) agents completed 10,000 episodes prematurely during the evaluation phase due to rapid task failure. Detailed analysis:

1. Reward, TSR: The (4:5:1) agent is optimal. This stems from its reduced Power Module proportion, which promotes aggressive tracking regardless of power cost. The TSR profiles of (4:4:2) and (4:3:3) agents proved comparable, indicating both ratios enable successful tracking.

2. PSR: The (4:4:2) agent outperformed others. As shown in **Appendix A.4**, while the (4:3:3) agent demonstrated superior PSR during training, it exhibited comparable PSR efficacy to (4:5:1) during evaluation. This equivalence arises from indirect compensation through exploration gains from increased ER.

3. ER: The (4:3:3) agent attained optimal. Its enhanced PSR compensated for ER constraints. The (4:5:1) agent surpassed the (4:4:2) agent in ER performance.

Table 2: Agent (4:4:2) is recommended for missions prioritizing persistent tracking. Agent (4:3:3) is optimal for tasks requiring concurrent tracking and spatial exploration. Agent (4:5:1) should be deployed for emergency tracking scenarios.

| $\omega_1 : \omega_2 : \omega_3$ | Rewards | TSR | PSR | ER |
|---|---|---|---|---|
| 4:4:2 | $1399.6034_{\pm 15.1}$ | $0.7979_{\pm 1.1\%}$ | $\mathbf{0.7031}_{\pm 1.5\%}$ | $0.2984_{\pm 1.1\%}$ |
| 4:2:4 | $917.5691_{\pm 7.5}$ | $0.5379_{\pm 2.2\%}$ | $0.4367_{\pm 1.1\%}$ | $0.1941_{\pm 1.2\%}$ |
| 4:3:3 | $1399.9049_{\pm 10.0}$ | $0.7886_{\pm 1.1\%}$ | $0.6374_{\pm 1.8\%}$ | $\mathbf{0.3440}_{\pm 2.2\%}$ |
| 4:5:1 | $\mathbf{1477.8414}_{\pm 4.5}$ | $\mathbf{0.8653}_{\pm 1.3\%}$ | $0.6581_{\pm 1.2\%}$ | $0.3307_{\pm 2.1\%}$ |
| 4:1:5 | $56.8749_{\pm 5.1}$ | $0.0334_{\pm 6.2\%}$ | $0.0239_{\pm 4.1\%}$ | $0.0118_{\pm 5.0\%}$ |

As presented in Table 2, these findings counterintuitively contradict our initial hypotheses. This divergence arises from the interdependence between PSR and ER. Specifically: The (4:4:2) agent achieved higher PSR and the (4:3:3) agent attained superior ER. Simulation experiments suggest that increased exploration distance reduces the probability of concurrent high-power maneuvers by both UAVs, thereby enhancing PSR. Conversely, greater power conservation decreases simultaneous acceleration events, which extends inter-UAV separation and improves ER.

As shown in Figure 1.b, the agent can be used in the task of continuous tracking (4:4:2), in the task of trend tracking and range exploration (4:3:3) and in the task of emergency tracking (4:5:1). Dynamic switching between these policy variants enables adaptation to complex tracking requirements.

## 6 RELATED WORKS

**Trajectory Prediction** (Zhao et al., 2018; Effati & Skonieczny, 2017) . **Target Detection and tracking** (Bochkovskiy et al., 2020; Duan et al., 2019; Bewley et al., 2016; Wojke et al., 2017; Wu et al., 2025a;b). **MARL Based tracking** (Zhou et al., 2022; Zhao et al., 2025b;a; Xia et al.,

2022; Chen et al., 2020; Moon et al., 2021): Compared with the aforementioned studies, our work addresses UAV swarm tracking in near-real-world scenarios, aiming to integrate tracking capability, power efficiency, exploration capacity, and communication effectiveness within a unified agent coordination framework. *Comparative table and settings in Appendix A.1*

## 7 CONCLUSION

To address practical engineering challenges including power efficiency, exploration, communication, and collision avoidance, we design a Coordinated Tracking Optimization Framework, enabling UAVs to accurately pursue targets while intelligently controlling exploration and power usage. Systematic simulations across small- and medium-scale target sets demonstrate strong scalability and stability. By adjusting reward components, the team exhibits multi-mode cooperative behaviors. We also perform preliminary studies in obstacle-present scenarios, aiming to further extend the algorithm's practical boundary.

## 8 REPRODUCIBILITY STATEMENT

Our reproducible **code** is available at https://anonymous.4open.science/r/CSE-ET. We provide several 2-vs-1 and 3-vs-1 scenario recordings in the **supplementary material**. The complete derivation of the spatial exploration reward is given in Appendix A.2. Training experiments for the multi-module reward are presented in Appendix A.4. A full description of the algorithm's pseudocode is provided in Appendix A.5. Figure 1 shows the simulation trajectories of CSE-ET, and trajectories for other algorithms are included in Figure 6. The simulator physics configuration follows Table 4, and the training setup follows Table 5.

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

# A APPENDIX

## A.1 RELATED WORKS

### A.1.1 FILTERING OR VISION BASED TRACKING

Trajectory Prediction: utilizing wireless sensor data from UAVs, target state prediction is achieved through filtering algorithms (e.g., Kalman Filter (Zhao et al., 2018), Extended Kalman Filter (Effati & Skonieczny, 2017)). However, non-stationarity in wireless channels frequently induces oscillatory motion in predicted trajectories.

Target Detection and tracking: modern detection frameworks (e.g., YOLOv4 (Bochkovskiy et al., 2020)/YOLOv7/CenterNet (Duan et al., 2019)) provide foundations for association-based trackers (SORT (Bewley et al., 2016)/ByteTrack/DeepSort (Wojke et al., 2017)). Subsequent research has contributed to enhanced tracking performance: Wu et al. (Wu et al., 2025a) employs a teacher-student model to augment visual feature extraction and pseudo-label bidding for target tracking in complex weather conditions, while Wu et al. (Wu et al., 2025b) introduces visual language model (VLM)-based perceptual prompts to improve tracking of targets across varying scales. Nevertheless, vision-based tracking methods predominantly adopt multi-stage pipelines (detection → association → coordinate transformation), which are computationally intensive and susceptible to error accumulation. Concurrently, video frame acquisition and transmission are constrained by physical hardware limitations, imposing significant constraints on practical deployment.

### A.1.2 MARL BASED TRACKING

Zhou et al. (Zhou et al., 2022) employ decentralized maximum reciprocal rewards to learn cooperative tracking policies for UAV swarms. Zhao et al. (Zhao et al., 2025b) utilize an attention-based communication protocol to enhance noise resilience in multi-UAV collaboration. Zhao et al. (Zhao et al., 2025a) propose a graph neural network (GNN)-based multi-agent reinforcement learning (MARL) algorithm to enrich environmental information and improve tracking performance. Xia et al. (Xia et al., 2022) introduce spatial information entropy to enable coordinated exploration among agents. Chen et al. (Chen et al., 2020) develop a feedback control mechanism addressing channel uncertainty to assist UAVs in locating RF targets. J. Moon et al. (Moon et al., 2021) present a deep reinforcement learning approach combining global and differential rewards to reduce tracking costs for UAVs. Compared with the aforementioned studies, our work addresses UAV swarm tracking in near-real-world scenarios, aiming to integrate tracking capability, power efficiency, exploration capacity, and communication effectiveness within a unified agent coordination framework.

## A.2 SPACE EXPLORATION RAWARD

### A.2.1 DERIVATION PROCESS OF K

When $0 \leq r_t \leq l$, the $\delta$ term contributes a constant 1 to the normalization (independent of $(r_t, \theta_t, \varphi_t)$), and thus $K$ is only a function of $l$:

$$\int_0^l \int_0^\pi \int_0^{2\pi} \zeta(r, \theta, \varphi, l) r^2 \sin\theta \, dr d\theta d\varphi = 1, \tag{30}$$

Plug in the definition and carry out term-wise integration:

$$1 = K(l) \left[ \int_0^l \int_0^\pi \int_0^{2\pi} e^{-\lambda r} r^2 \sin\theta \, dr d\theta d\varphi + \frac{1}{\lambda} \int \int \int \delta_{r,\theta,\varphi}(\cdot) r^2 \sin\theta \, dr d\theta d\varphi \right] \tag{31}$$

$$= K(l) \left[ 4\pi \int_0^l r^2 e^{-\lambda r} dr + \frac{1}{\lambda} \right], \tag{32}$$

The expression of $K$ is obtained as follows:

$$K = \frac{\lambda}{4\pi\lambda\kappa(l) + 1}, \quad \kappa(l) := \int_0^l r^2 e^{-\lambda r} dr. \tag{33}$$

Calculate indefinite integral $\kappa(l)$:

$$\int r^2 e^{-\lambda r} dr = e^{-\lambda r} \left( -\frac{r^2}{\lambda} - \frac{2r}{\lambda^2} - \frac{2}{\lambda^3} \right) + C, \tag{34}$$

$$\kappa(l) = \frac{1}{\lambda^3} \left[ 2 - e^{-\lambda l} \left( \lambda^2 l^2 + 2\lambda l + 2 \right) \right] \tag{35}$$

Finally, the formula of $K$ is obtained as follows:

$$K = \frac{\lambda}{1 + \frac{4\pi}{\lambda^2} \left[ 2 - e^{-\lambda l} \left( \lambda^2 l^2 + 2\lambda l + 2 \right) \right]}. \tag{36}$$

### A.2.2 DERIVATION PROCESS OF SPATIAL INFORMATION ENTROPY

Similar to information entropy, the spatial information entropy corresponding to information correlation density is defined as:

$$\mathcal{H}(l) = - \int_0^\pi \int_0^l \int_0^{2\pi} r^2 \zeta(r, \theta, \varphi, l) \ln \zeta(r, \theta, \varphi, l) \, dr d\theta d\varphi. \tag{37}$$

Divide $\zeta(r, \theta, \varphi, l)$ into continuous part $I_c = K e^{-\lambda r}$ and pulse part $I_\delta = \frac{K}{\lambda} \delta$, and record $d\Omega = \sin\theta d\theta d\varphi$ to get:

$$I_c = -K \int r^2 e^{-\lambda r} (\ln K - \lambda r) d\Omega dr \tag{38}$$

$$= -4\pi K \left[ \ln K \int_0^l r^2 e^{-\lambda r} dr - \lambda \int_0^l r^3 e^{-\lambda r} dr \right] \tag{39}$$

$$= -4\pi K \left[ \ln K \int_0^l r^2 e^{-\lambda r} dr - \lambda \left( -\frac{r^3 e^{-\lambda r}}{\lambda} + \frac{3}{\lambda} \int r^2 e^{-\lambda r} dr \right) \right] \tag{40}$$

$$= 4\pi K \left[ (3 - \ln K)\kappa(l) - l^3 e^{-\lambda l} \right]. \tag{41}$$

$$\tag{42}$$

For the derivation of the pulse part, narrow kernel regularization is adopted, and its limit is given only by $\ln K - \lambda r_t$, which is represented by symbol $h(\cdot)$:

$$I_\delta = K h (\ln K - \lambda r_t). \tag{43}$$

Obtained by algebraic calculation:

$$\mathcal{H}(l) = 4\pi K(l) \left[ (3 - \ln K(l)) \kappa(l) - l^3 e^{-\lambda l} \right] + K(l) h (\ln K(l) - \lambda r_t). \tag{44}$$

Since $\kappa'(l) = l^2 e^{-\lambda l} > 0$ and $\kappa(l)$ increases monotonically with $l$, $K(l)$ decreases monotonically with $l$ and converges to a finite limit. Finally, we obtain that $\mathcal{H}(l)$ increases monotonically with $l$ and then saturates, which accords with the intuition that, during exploration, information acquired by different UAVs evolves from overlapping to fully independent.

### A.3 TRAINING SETTINGS

Based on the Isaac Gym platform, the simulation environment offers two primary advantages:

1. Physics simulations within Isaac Gym execute on the GPU, with results residing in PyTorch GPU tensors.

2. Utilizing Isaac Gym's tensor-based APIs enables the computation of observations and rewards directly on PyTorch's GPU, thus facilitating the parallel execution of thousands of environments on a single workstation.

Owing to the well-designed reward function and the platform's acceleration capabilities, training time on a single NVIDIA GeForce RTX 3090 GPU is reduced to approximately 10 hours. With

the number of parallel environments set to 2048, this approach enhances training efficiency while conserving computational resources (Makoviychuk et al., 2021).

"Batch Size" refers to the number of environmental steps collected before updating the policy via gradient descent. "Mini Batch Size" specifies the number of subdivisions within a batch of data. "Gain" denotes the weight initialization scaling factor applied to the actor network's final layer. "Gamma" represents the discount factor in reinforcement learning. "Clip Param" is the hyperparameter controlling the clipping threshold in policy objective and value loss functions, which penalizes excessive policy and value function updates.

The network architecture is structured as follows: It begins with "Num GRU Layers" layers, each with dimensionality defined by "RNN Hidden State Dim", followed by an MLP network comprising "num fc" fully-connected layers. The dimension of each fully-connected layer is uniformly specified by the 'FC Layer Dim' hyperparameter. Detailed parameter settings are provided in Table 5.

### A.4 SUPPLEMENTARY EXPERIMENT

As shown in Figure 4, the agent (4:1:5) and (4:2:4) failed to accomplish the tracking task due to proportional imbalance, resulting in training failure. Among the PSR, the (4:4:2) demonstrated optimal performance.

For Reward, TSR, and PSR, the training curves of (4:4:2), (4:3:3), and (4:5:1) exhibited close alignment. This convergence arises from cross-component interference within the model's architecture. The conclusions presented in the main text derive from integrated analysis of training outcomes and evaluation results. As illustrated in Fig. other-tail, we visualize the positional distribution and velocity vectors of the UAV swarm. The divergent escape trajectories of target UAVs per episode induce dynamic spatial distribution fluctuations. Trajectories employ color gradients to represent velocity magnitude, where brighter hues on the colorbar indicate higher speeds. Escape UAVs are expressed in the opposite way.

Agents trained with all three algorithms successfully accomplish fundamental tracking tasks. They maintain effective tracking against diverse evasion maneuvers—including linear escape, vertical climbs, diving actions, and coordinated turns—executed by target UAV. Crucially, the tracking swarm demonstrates robust tracking stability even during high-agility evasion scenarios involving abrupt directional changes.

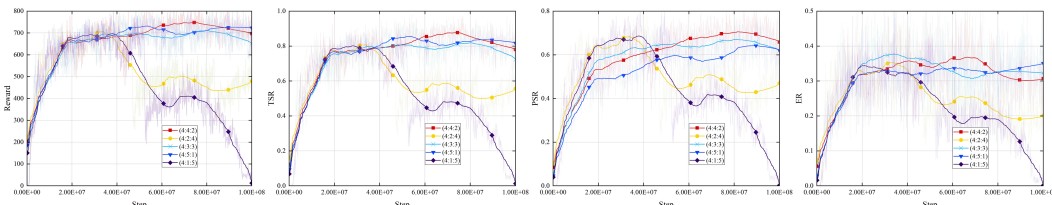

Figure 4: Training Comparison of Five Proportional Parameters

### A.5 ALGORITHM DETAILS

The generalized advantage estimator GAE (Schulman et al., 2015) is used to get $A_\pi(s, \mathbf{a})$,

$$\hat{A}_i(t) = \zeta_i(t) + (\gamma\varrho)\zeta_i(t+1) + \cdots + (\gamma\varrho)^{T-t+1}\zeta_i(T-1), \tag{45}$$

where $\zeta_i(t) = R(t) + \gamma V_\phi(s_{t+1}) - V_\phi(s_t)$, $\varrho$ is the GAE discount factor, and the value of the value function is expressed as $\hat{V}_\phi(s_t) = \hat{A}_i(t) + V_\phi(s_{t+1})$. It can be further normalized using PopArt (Hessel et al., 2019):

$$\hat{\mu} = (1-\delta)\mu + \delta \sum_{i=1}^{N} \sum_{t=1}^{T} \hat{V}_\phi(s_t)/BN, \tag{46a}$$

$$\hat{v} = (1-\delta)v + \delta \sum_{i=1}^{N} \sum_{t=1}^{T} (\hat{V}_\phi(s_t))^2/BN, \tag{46b}$$

where $\hat{\sigma}$ is the smoothing parameter. According to standard deviation $\hat{\sigma} = \sqrt{\hat{\nu} - \hat{\mu}^2}$, the normalized value function is obtained as $\tilde{V}_\phi(s_t) = \left(\hat{V}_\phi(s_t) - \hat{\mu}\right)/\hat{\sigma}$.

---

**Algorithm 1** CSE-ET

---

**Input:** Learning rate $\alpha$, Environment numbers $B$, Number of UAVs: $N$ per environment , Data chunks of length $L$, Steps per episode $T$.
**Initialize:** $\forall n \in N, \forall i \in B$, Actor networks $\{\theta_0^{i_n}\}$, Global V-value network $\{\phi_0^i\}$
**while** $step \leq step_{\max}$ **do**
    set data buffer $\mathcal{D} = \{\}$
    **for** $i = 1$ **to** $B$ **do**
        $\Gamma = []$ empty list
        initialize $h_{0,\pi}^{(i_1)}, \ldots h_{0,\pi}^{(i_n)}$ actor RNN states
        initialize $h_{0,V}^{(i_1)}$ critic RNN states
        **for** $t = 1$ **to** $T$ **do**
            **for** UAV $n = 1$ **to** $N$ **do**
            $p_t^{(i_n)}, h_{t,\pi}^{(i_n)} = \pi(o_t^{(i_n)}, h_{t-1,\pi}^{(i_n)}; \theta)$
            $a_t^{(i_n)} \sim p_t^{(i_n)}$
            $v_t^{(i_n)}, h_{t,V}^{(i_1)} = V(s_t^{(i_n)}, h_{t-1,V}^{(i_1)}; \phi)$
            **end for**
            $\Gamma + = [s_t, o_t, h_{t,\pi}, h_{t,V}, a_t, r_t, s_{t+1}, o_{t+1}]$
        **end for**
        Compute $\hat{A}$ via GAE on $\Gamma$, using PopArt
        Compute $\hat{R}$ by PopArt on $\Gamma$
        **for** $l = 0$ **to** $T//L$ **do**
            $\mathcal{D} = \mathcal{D} \cup (\Gamma[l : l+T], \hat{A}[l : l+L], \hat{R}[l : l+L])$
        **end for**
    **end for**
    **for** mini-batch k = 1,...,$K$ **do**
        $b \leftarrow$ simple random mini-batch from $\mathcal{D}$
        Collect a set of trajectories $\pi_{\theta_k} = (\pi_{\theta_k^1}^1, \ldots, \pi_{\theta_k^n}^n)$.
        Random ordering agents $i_{1:n}$
        Set $M^{i_1}(s, \mathbf{a}) = \hat{A}(s, \mathbf{a})$
        **for** $i_m = i_1, \ldots, i_n$ **do**
            Adam update $\theta^{i_m}$ on $L(\theta^{i_m})$ with data b
            Compute $M^{i_{1:m+1}}$ with $\theta_{k+1}^{i_m}$
        **end for**
        Adam update $\phi$ on $L(\phi)$ with data b
    **end for**
  **edn while**

---

## A.6 ABLATION EXPERIMENT

Table 3: Final evaluation results of progressive techniques applied to CSE-ET

| Model | Rewards | TSR | PSR | ER |
|---|---|---|---|---|
| CSE-ET | **1399.9049**$_{\pm10.0}$ | **0.7886**$_{\pm1.1\%}$ | **0.6374**$_{\pm1.8\%}$ | 0.3440$_{\pm2.2\%}$ |
| MAPPO | 1302.7120$_{\pm14.0}$ | 0.7188$_{\pm1.2\%}$ | 0.5916$_{\pm1.3\%}$ | **0.3703**$_{\pm1.2\%}$ |
| HAPPO | 1094.2966$_{\pm6.1}$ | 0.5874$_{\pm1.1\%}$ | 0.4506$_{\pm1.2\%}$ | 0.2450$_{\pm1.2\%}$ |
| HATRPO | 1163.4772$_{\pm3.4}$ | 0.6253$_{\pm1.6\%}$ | 0.5001$_{\pm1.2\%}$ | 0.2448$_{\pm1.3\%}$ |
| Two Stage | 576.3578$_{\pm17.7}$ | 0.0216$_{\pm2.2\%}$ | 0.0246$_{\pm2.4\%}$ | 0.0059$_{\pm1.6\%}$ |
| Observation Normalization | $-15.7531_{\pm20.6}$ | 0.003$_{\pm0.1\%}$ | 0.003$_{\pm0.15\%}$ | 0.001$_{\pm0.1\%}$ |

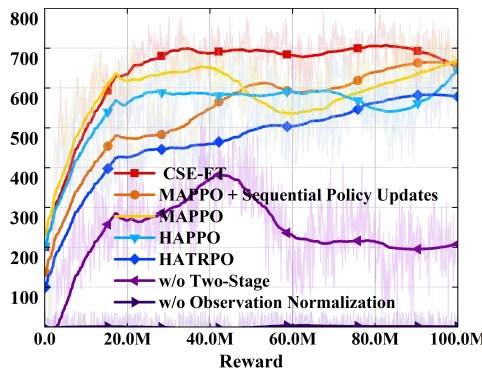 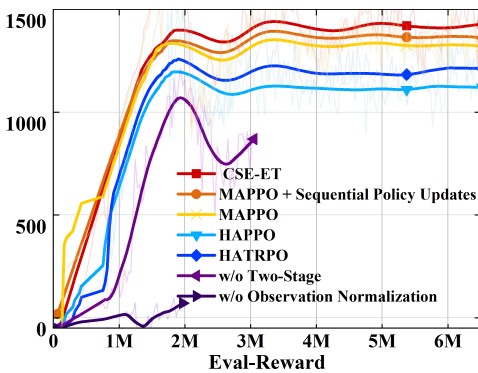

Figure 5: Training and evaluation results of progressive techniques applied to CSE-ET

We conduct ablation studies on the key components of the algorithm to further clarify their contributions to performance. PopArt normalizes the value-function targets across different tasks (pursuit, exploration, energy saving) and different phases (Phase I: accelerated pursuit; Phase II: steady pursuit; plus learning exploration and energy-saving behaviors). It is a widely validated, general-purpose component and is kept with a fixed configuration in all experiments below.

As shown in Fig. 5, we evaluate the components that drive performance in both the training and evaluation phases, including: MAPPO without Two-Stage, MAPPO without observation normalization, HATRPO, HAPPO, MAPPO, and MAPPO + Sequential Policy Updates. Based on the final evaluation results in Table 3, we conclude:

1. Observation normalization is crucial for stable convergence; two-stage curriculum learning ensures the acquisition of post-pursuit behaviors such as energy management and exploration. Removing either component leads to task failure.

2. At 60 million steps, MAPPO suffers a training collapse caused by policy conflicts during training, which degrades performance. MAPPO + Sequential Policy Updates produces a smoother training curve and outperforms pure MAPPO in the final evaluation.

3. Building on Sequential Policy Updates, CSE-ET further incorporates an RNN, which improves performance and achieves the best results on both the training and evaluation curves.

## A.7 LLMs USAGE STATEMENT

Throughout manuscript preparation, we used large language models solely for editorial copy-editing and formatting—e.g., spelling, grammar, punctuation, diction, and flow. We did not employ them for problem selection or framing, content generation, literature synthesis, data/figure generation or analysis, experimental design, result interpretation, or drawing conclusions. No substantive scholarly contribution relied on LLMs.

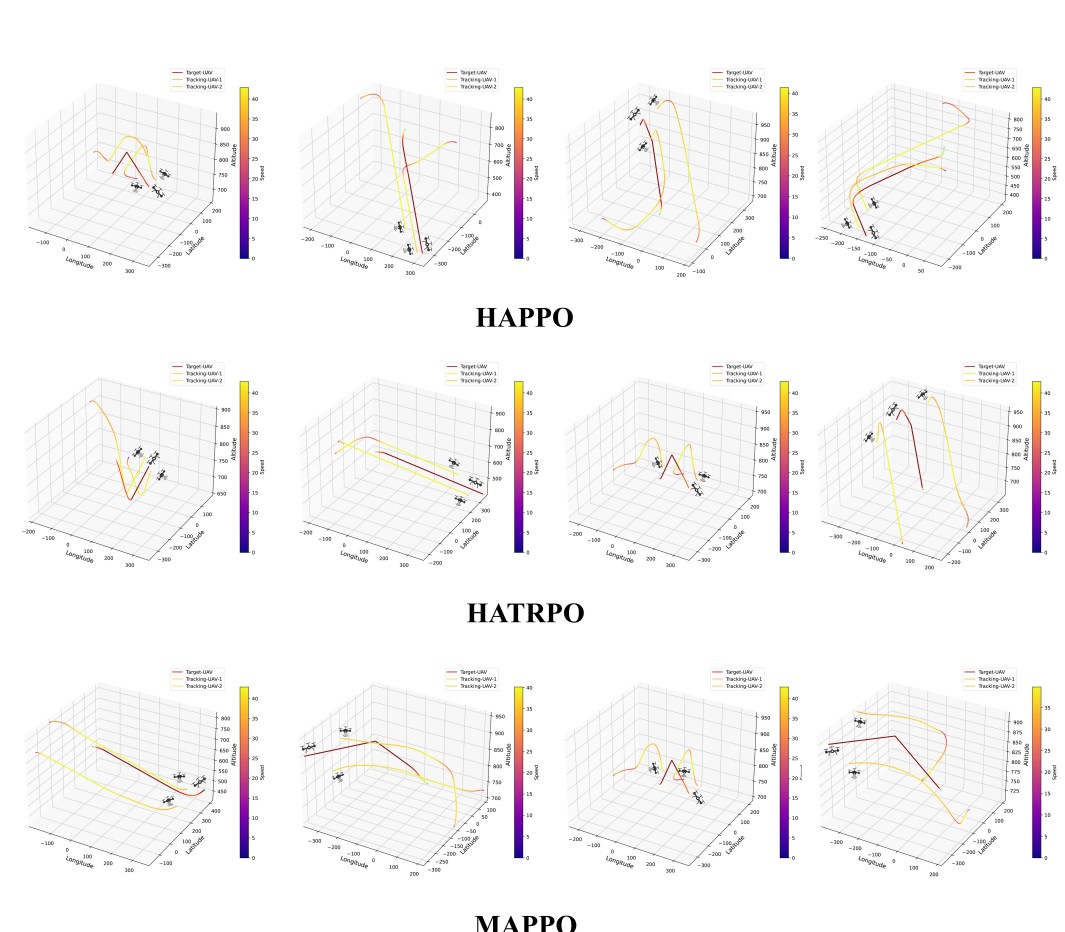

Figure 6: Simulation performance of HAPPO, HATRPO and MAPPO

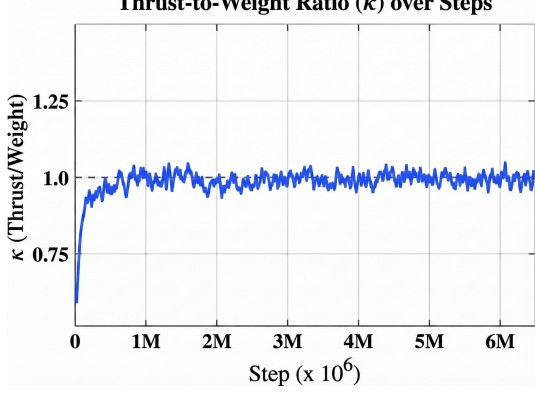

Figure 7: Thrust-to-weight ratio ($\kappa$) of each step in CSE-ET training process

Table 4: Physical parameters of simulation environment

| Notation | Physical Meaning | Value |
|---|---|---|
| $H$ | Flight Altitude | $750m$ |
| $\Delta t$ | Time Interval | $0.02s$ |
| $P_e$ | Initial Position of the Escape UAV | $(0, 0, H)$ |
| $P_i$ | Initial Position of the UAVs | $x^2 + y^2 \in (220^2, 250^2),$ $\|z - H\| \le 150$ |
| $V_e$ | Initial Velocity of the Escape UAV | $0m^2/s$ |
| $V_i$ | Initial Velocity of the UAVs | $\|V_i\| \le 42.87m/s$ |
| $N$ | Number of UAVs | 2 |
| $L$ | Sliding Window Size | 5 |
| $d_{\min}^{ut}$ | Escape-Pursuit Safe Distance | $50m$ |
| $d_{\min}^{uu}$ | UAV-UAV Safe Distance | $25m$ |
| $\rho_s$ | Range of tracking | $200m$ |
| $\xi_v$ | Velocity Constraint | $42.87m/s$ |
| $\xi_a$ | Acceleration Constraint | $50m^2/s$ |
| $\delta$ | Channel Fading Factor | 2 |
| $\sigma^2$ | Noise Power | $-43dBm$ |
| $G$ | Power Gain | 1 |
| $P$ | Transmission Power | $10dBm$ |
| $\zeta_{th}$ | Data Rate Threshold | $1.218bps/Hz$ |
| $\hat{d}[\hat{H}]$ | Equivalent $\mathcal{H}(l)$ Distance | $150m$ |
| $A$ | Rotor Discarea | $0.503m^2$ |
| $\tau$ | Rotor Solidity | 0.05 |
| $\varphi$ | Fuselage Drag Ratio | 0.6 |
| $U_{tip}$ | Tip Speed | 120 |
| $\rho$ | Air Density | $1.225kg/m^3$ |

Table 5: Training parameters

| Hyperparameters | Value |
|---|---|
| Parallel environments | 2048 |
| Speed up Episode Steps | 400 |
| Max Consecutive Steps | 600 |
| Total Environmental Steps | 100000000 |
| Data Chunk Length | 16384 |
| Batch Size | $N_{\text{envs}} \times L_{\text{buf}} \times N_{\text{agents}}$ |
| Mini Batch Size | batch size / mini-batch |
| Gamma | 0.96 |
| Gae Lambda | 0.95 |
| Huber Delta | 10.0 |
| Clip Param | 0.2 |
| PPO Epoch | 5 |
| Optimizer | Adam |
| Optimizer Epsilon | 1.e-4 |
| Gain | 0.01 |
| Num GRU Layers | 1 |
| RNN Hidden State Dim | 1024 |
| FC Layer Dim | 1024, 512 |
| Num FC | 2 |
| Network Initialization | Orthogonal |
| Use Reward Normalization | True |
| Use Feature Normalization | True |

