# OpenReview forum: "CSE-ET: Joint Optimization for 3D Multi-UAV Explore-and-Track under Connectivity, Separation, and Energy Constraints"
_ICLR.cc/2026/Conference — ICLR 2026 Conference Withdrawn Submission_

### Official Review · Reviewer_uAAm · 2025-10-30

**Soundness:** 1
**Presentation:** 2
**Contribution:** 2
**Rating:** 2
**Confidence:** 3

**Summary:**

The paper tackles the problem of targeting an evading UAV, with an optimal avoidance policy, by jointly considering communication,
spatial exploration, power consumption, and inter-UAV collision avoidance for the tracking team. It proposes to design a unified joint-optimization framework based on multi-agent reinforcement learning is ambitious.

**Strengths:**

The problem of targeting an evading UAV has been tacked using a holistic approach considering communication,
spatial exploration, power consumption, and inter-UAV collision avoidance.

**Weaknesses:**

The authors claim in the abstract that ‘To our knowledge, this paper presents the 3D multi-UAV cooperative tracking framework that is most closely aligned with practical constraints.’, But, the system model considered by the authors present serious problems which make their problem unrealistic.

For instance, the model used (1a-1c) supposedly represents the UAV. The problem is that according to Fig. 1 they are considering a quadrotor which has  4 different variables to control it; this allow to control the position at x,y and z independently. But the model (1a-1c) has only two angles to control the 3 velocities (x,y,z). This implies that they cannot control all 3 velocities independently which means that this model does not seem to accurately describe the motion of the UAV considered in the paper.
Furthermore, the choice for the angles in model   (1a-1c) seems unortodox. Usually the orientation is described with the Euler angles (roll, pitch and yaw) or with quaternions.

 In (2), the authors present an arbitrary model for the Evader, without any support of the literature. This model seems arbitrary and heuristic, but it also seems to disregard the rich body of literature on the problem of Hunter-Prey which present already sophisticated equations for the behaviour of the Evader agent.

In (4), the authors are comparing an SNR with a distance, which is inconsistent in terms of the physical units.

The power consumption model (6) presented by the authors cannot be used for the scenario considered by them. The reason for this is that in (Zeng et al 2019), the original authors mention that propose such power consumption model, they state that V is the forward velocity, impliying a 2D motion (i.e. no velocity on the z axis). But the authors of the paper under evaluation assume a 3D motion which implies a z-axis velocity which is not supported by the power consumption model used by the authors.

The reward expressions have been introduced without justification or citing references.

**Questions:**

See points raised in the Weaknesses section

---

> ### Author Response · Authors · 2025-11-23
>
> # Q1
> In our model, ϕb(t) and φb(t) are not the UAV’s attitude angles (e.g., roll/pitch/yaw in Euler angles) and do not attempt to fully describe the quadrotor’s rigid-body attitude. They are used solely to parameterize the direction of the acceleration vector a(t) in the inertial frame, i.e., a spherical-coordinate–like direction parameterization. Therefore, ${\phi_b}(t)$ and ${\varphi_b}(t)$ only specify the acceleration direction, while the magnitude of the acceleration is controlled separately by **a(t)**. The angle definitions are as follows:
> 1. The angle ${\phi_b}(t)$ denotes the body angle between the velocity vector and the $o\text{-}x\text{-}y$ plane.
> 2. The angle ${\varphi_b}(t)$ represents the angle between the projection of the velocity vector $v'$ onto the $o\text{-}x\text{-}y$ plane and the $o\text{-}y$ axis.
>
> Accordingly, the acceleration vector can be written as:
>
> $a_x(t) = a(t)\cos\phi_b(t)\sin\varphi_b(t)$
>
> $a_y(t) = a(t)\cos\phi_b(t)\cos\varphi_b(t)$
>
> $a_z(t) = a(t)\sin\phi_b(t)$
>
> # Q2
> We thank the reviewers for pointing out the extensive body of work on evasion behavior in the pursuit–evasion problem. Indeed, prior literature based on differential games or reinforcement learning can derive more complex—even optimal—evasion laws.
>
> However, our goal is not to propose a new optimal evasion model, but to adopt a simple, tunable, and physically reasonable evasion policy for constructing multi-UAV pursuit–evasion scenarios, so that we can focus on evaluating our collaborative tracking joint optimization framework that integrates exploration, energy saving, and communication. Our evasion strategy is supported by existing literature:
>
> Specifically, in Eq. (2),
> $D_{\alpha}(t)=\sum_{l=0}^{L} \alpha^{l} P_e(t-l)$
> is an exponentially weighted moving average (EWMA) of the evader’s past positions—a standard sliding-window smoothing technique—used to balance “responsive reaction” against “trajectory smoothness.” Similar issues in time-series smoothing and window-length selection have been systematically discussed for UAV-collected trajectory data, e.g., (Ravikumar et al., 2025; Thiemann et al., 2008;). A shorter window length $L$ or a larger discount factor $\alpha$ produces more agile evasion, whereas a larger $L$ or a smaller $\alpha$ suppresses high-frequency maneuvers and increases trajectory smoothness. As shown in Fig. 2.a, when we vary the responsiveness of the evasion model and test tracking performance, our method lags behind MAPPO only on a single ER metric at $L=5$; in all other cases, it outperforms the baselines. To validate the robustness of our approach, the experiments in Fig. 3 are conducted under $L=5$.
>
> On this basis, we construct the evasion acceleration components using the line-of-sight directions between the smoothed position $D_{\alpha}(t)$ and each pursuer UAV $P_i(t-1)$; the resultant always points away from the pursuers. This construction is consistent with the classical pure evasion strategy (i.e., the evader moves in the direction opposite to the line to the pursuer) (Von Moll et al., 2020; Isaacs, 1965) , except that in the multi-pursuer case we combine the “repulsive forces” from different pursuers via linear superposition / artificial potential fields.
>
> Therefore, we acknowledge that Eq. (2) is a heuristic kinematic control law, but it is not ad hoc; it is a lightweight implementation that integrates three established ideas—trajectory smoothing, line-of-sight geometry, and artificial potential fields. To avoid misunderstanding, we will add the above references and clarify the role of this model in the revised manuscript.
>
> Reference:
>
> - Ravikumar, S. H., Maurya, A. K., & Arkatkar, S., Evaluation of Smoothing Techniques for Vehicular Trajectory Data from UAVs, in Proc. 7th Int. Conf. of Transportation Research Group of India (CTRG 2023), LNCE 426, Springer, 2025.
> - Thiemann, C., Treiber, M., & Kesting, A., "Estimating Acceleration and Lane-Changing Dynamics from Next Generation Simulation Trajectory Data," Transportation Research Record, vol. 2088, pp. 90–101, 2008.
> - Isaacs, R., Differential Games: A Mathematical Theory with Applications to Warfare and Pursuit, Control and Optimization,
> - Wiley, 1965. Von Moll, A., Fuchs, Z., & Pachter, M., "Optimal Evasion Against Dual Pure Pursuit," Proc. American Control Conference (ACC), 2020.

---

> ### Author Response · Authors · 2025-11-23
>
> # Q3
> We appreciate the reviewer’s correction. We agree that our original wording about the threshold was not rigorous and could indeed be misconstrued as “directly comparing SNR with distance.” Here we clarify the actual meaning of the model and will correct the related notation and descriptions in the revision.
>
>
> First, we use a linear-scale (non-dB) SNR model:
>
> $\mathrm{SNR}{ij}(t)=\frac{GP}{\sigma^2}d{ij}(t)^{-\delta}$,
>
> where $G$ is a dimensionless gain, and $P$ and $\sigma^{2}$ are the transmit power and noise power (with the same units), respectively; $\delta$ is the path-loss exponent. Thus $\mathrm{SNR}_{ij}$ is a dimensionless power ratio and is physically self-consistent in terms of units.
>
> According to the Shannon formula, to support the target spectral efficiency $\zeta_{\mathrm{th}}$ (bit/s/Hz), the link must satisfy $\mathrm{SNR}{ij}(t)\geq\gamma{\mathrm{th}}$,$\quad\gamma_{\mathrm{th}}=2^{\zeta_{\mathrm{th}}}-1$,
> where $\gamma_{\mathrm{th}}$ is likewise a dimensionless SNR threshold. Substituting this into the path-loss model yields an equivalent constraint in terms of distance:
>
> $\mathrm{SNR}{ij}(t)\geq\gamma{\mathrm{th}},\quad\gamma_{\mathrm{th}}=2^{\zeta_{\mathrm{th}}}-1$,
>
> with
> $\beta=\left(\frac{GP}{\sigma^2\gamma_{\mathrm{th}}}\right)^{1/\delta}$
> being the maximum communication-distance threshold derived from the SNR threshold. Its unit is identical to that of $d_{ij}$ (both are distances).
>
> In implementation, we therefore write the link weight as
> $\beta=\left(\frac{GP}{\sigma^2\gamma_{\mathrm{th}}}\right)^{1/\delta}$
> i.e., when $\mathrm{SNR}\_{ij}(t)\ge\gamma\_{\mathrm{th}}$ (equivalently $d\_{ij}(t)\le\beta$), we use an SNR-based weight; otherwise the weight is set to zero. Hence, the model actually compares SNR against an SNR threshold—or distance against a distance threshold—and never “directly compares SNR with distance” in physical units.
>
> We acknowledge that the sentence “when $a_{ij}(t)$ falls below a predefined threshold $\beta$” in the original manuscript is misleading in notation (it appears to compare $a_{ij}$ with $\beta$). This will be corrected in the revision, for example, to:
> “the link is considered connected when $\mathrm{SNR}\_{ij}(t)\ge\gamma\_{\mathrm{th}}$ (equivalently $d\_{ij}(t)\le\beta$).”
>
> We will also state explicitly at the appropriate place that $\beta$ is the maximum communication-distance threshold. This makes the unit consistency and the connectivity criterion clearer.

---

> ### Author Response · Authors · 2025-11-23
>
> # Q4
> We thank the reviewer for the remark. We agree that Eq. (6) is a simplification from a more general model under certain flight assumptions, and its applicability should be stated. Below we clarify from two perspectives: the theoretical model and our specific reinforcement learning implementation.
>
> **1. On the origin and assumptions of the model by Zeng et al.**
>
> Specifically, in Appendix A of Zeng et al., the authors first derive a general power model $P(V,\kappa)$ that depends on flight speed $V$ and the thrust-to-weight ratio $\kappa=T/W$ (see their Eq. (66)). In this model, $V$ denotes the UAV’s speed magnitude, while $\kappa$ captures the ratio of thrust to weight. Next, using the vertical force balance $T\cos\alpha_T \approx W$, they obtain $\kappa\approx1$ and simplify $P(V,\kappa)$ to a function of the speed magnitude $V$ only—this is the form we cite as Eq. (6). This treatment is intended to yield a closed-form, tractable power model for communication-oriented optimization, and the original text explicitly ignores acceleration-related energy costs.
>
> **2. Applicability in 3D from an RL perspective: at fine time steps, a local approximation $\kappa\approx1$**
>
> Our 3D trajectories are produced via agent–environment interaction in reinforcement learning. We adopt a time step of $\Delta t = 0.01s$. At each discrete time step:
> - the agent selects an action based on the current state;
> - the environment updates the UAV’s position and velocity and uses Eq. (6) to compute propulsion energy for that step, yielding the immediate reward;
> - explicit constraints on speed and acceleration are imposed in the action space to ensure temporal smoothness of the trajectory.
>
> Hence, while the UAV’s altitude can exhibit “large” changes globally, these arise from the accumulation of many tiny climbs/descents across steps of $0.01\,\text{s}$. From a differential/discrete-time viewpoint, within each single step:
> - the flight path angle changes only slightly relative to the previous instant;
> - the deviation of the required thrust-to-weight ratio $\kappa$ from $1$ is also small.
>
> As in **Fig. 7**, this is equivalent to locally linearizing $P(V,\kappa)$ around $\kappa=1$ and truncating higher-order terms. Under such fine-grained discretization ($\Delta t=0.01s$) and speed/acceleration constraints, using $P(V)$ (Eq. (6))—which depends only on speed magnitude—to compute per-step propulsion power yields a per-step approximation error that is a higher-order small quantity relative to the UAV’s total energy; its cumulative effect is much smaller than channel randomness and other uncertainties. Within this implementation framework, in the 3D setting we interpret the $v$ in Eq. (6) as the UAV’s airspeed magnitude $\||{\bf v(t)}\||\$, and we operationalize the original assumption “$\kappa\approx1$” at each local time step.
>
> **3. Research goal and notes on model applicability**
>
> Our contribution is a stable training framework for “strong partial observability + resource constraints” (with a unified objective and constraints spanning pursuit/exploration/energy saving/communication/collaboration), rather than high-fidelity aerodynamic modeling of every extreme attitude. Accordingly, we follow the simplified power model widely used in the UAV communications literature and, combined with RL’s stepwise decision-making, the fine time discretization $\Delta t=0.01s$, and speed/acceleration constraints, it remains suitable in 3D scenarios.
>
> In the revision, we have added “Assumptions and applicability of the power model,” stating explicitly:
>
> “This paper adopts the closed-form propulsion power model for rotary-wing UAVs given in [Zeng]. The model is simplified from the more general $P(V,\kappa)$ under the assumptions of straight, near-level flight and a thrust-to-weight ratio close to 1. In this work, the UAV executes smooth 3D trajectories under velocity and acceleration constraints in the action space with a time step of $\Delta t=0.01s$. The experiments in Fig. 7 indicate that Eq. (6) is a reasonable engineering approximation for characterizing instantaneous propulsion power.
>  For extreme maneuvers such as aggressive climb/dive, incorporating the full $\kappa$-dependent model or additional potential-energy terms is a meaningful extension, but beyond the scope of this paper.”

---

> ### Author Response · Authors · 2025-11-23
>
> # Q5
> We appreciate the reviewer’s comment. The reward function consists of three parts:
>
> • TRACKING REWARD: We explain the tracking advantage and its meaning; details are provided in Table 1 of Sec. 3.1 (STATE SPACE), with additional description in Table 3 of the Appendix. A schematic illustration is given in Fig. 1a.
>
> • SPACE EXPLORATION REWARD: We describe the key steps for computing spatial information entropy in this section; the full derivation is provided in Appendix A.2.
>
> • POWER CONSUMPTION REWARD: The definition in this section is built upon the energy-consumption model in Sec. 2.4.
>
> In summary, we have provided explanations and derivations for all components of the reward function, and we will add further descriptive details in the revision.

---

### Official Review · Reviewer_SGWU · 2025-10-31

**Soundness:** 3
**Presentation:** 3
**Contribution:** 3
**Rating:** 6
**Confidence:** 4

**Summary:**

This paper presents a comprehensive multi-agent reinforcement learning (MARL) framework for solving practical and complex multi-target tracking problems using UAV swarms. Despite making very strong contributions, several potential weaknesses exist with respect to the proposed "sequential policy update" mechanism, which requires further analysis.

**Strengths:**

The paper's biggest contribution is that it clearly diagnosed the instability experienced by standard MARL algorithms (MAPPOs) in complex and conflicting multi-compensation (tracking, exploration, power) environments. The authors show that this instability is attributed to a "policy conflict" and present a unique and effective solution called "sequential policy update (CSE-ET) to address it.

The paper is very clearly and systematically described in three stages: 'Base Model', 'Decision Model' (MARL Definition) and 'Optimization Algorithm'. In particular, the overall framework, Figures 2 and 3, etc. greatly help the reader intuitively understand the core ideas and their effects of the proposed technology.

By setting strong and appropriate SOTA MARL algorithms such as MAPPO, HAPPO, and HATRPO as comparators, we quantitatively demonstrate the superiority (tracking success rate, power saving, etc.) of the proposed CSE-ET. This robust experimental design gives high confidence in the performance improvement results of the proposed framework.

**Weaknesses:**

The paper's biggest contribution is that it clearly diagnosed the instability experienced by standard MARL algorithms (MAPPOs) in complex and conflicting multi-compensation (tracking, exploration, power) environments. The authors show that this instability is attributed to a "policy conflict" and present a unique and effective solution called "sequential policy update (CSE-ET) to address it.

The paper is very clearly and systematically described in three stages: 'Base Model', 'Decision Model' (MARL Definition) and 'Optimization Algorithm'. In particular, the overall framework, Figures 2 and 3, etc. greatly help the reader intuitively understand the core ideas and their effects of the proposed technology.

By setting strong and appropriate SOTA MARL algorithms such as MAPPO, HAPPO, and HATRPO as comparators, we quantitatively demonstrate the superiority (tracking success rate, power saving, etc.) of the proposed CSE-ET. This robust experimental design gives high confidence in the performance improvement results of the proposed framework.

**Questions:**

The paper's biggest contribution is that it clearly diagnosed the instability experienced by standard MARL algorithms (MAPPOs) in complex and conflicting multi-compensation (tracking, exploration, power) environments. The authors show that this instability is attributed to a "policy conflict" and present a unique and effective solution called "sequential policy update (CSE-ET) to address it.

The paper is very clearly and systematically described in three stages: 'Base Model', 'Decision Model' (MARL Definition) and 'Optimization Algorithm'. In particular, the overall framework, Figures 2 and 3, etc. greatly help the reader intuitively understand the core ideas and their effects of the proposed technology.

By setting strong and appropriate SOTA MARL algorithms such as MAPPO, HAPPO, and HATRPO as comparators, we quantitatively demonstrate the superiority (tracking success rate, power saving, etc.) of the proposed CSE-ET. This robust experimental design gives high confidence in the performance improvement results of the proposed framework.

---

> ### Author Response · Authors · 2025-11-24
>
> We sincerely thank the reviewers for their recognition and positive evaluation of our work. We are also very grateful for their concrete acknowledgement of this paper’s contributions, for example:
> 1) The reviewers recognize that our learning framework for constrained cooperative target tracking in open airspace couples spatial information-entropy-driven active exploration and energy efficiency with the tracking process, and can be trained stably in resource-constrained, partially observable multi-agent decision-making scenarios.
> 2) The reviewers praise that the paper describes the three stages—“basic model,” “decision model” (MARL formulation), and “optimization algorithm”—in a very clear and systematic manner. In particular, the overall framework and Figures 2 and 3 greatly help readers intuitively understand the core ideas of the proposed techniques and their effects.
> 3) The reviewers find that our experiments provide a comprehensive evaluation and systematic ablations against multi-agent baselines MAPPO, HAPPO, and HATRPO, and the results demonstrate the superiority of the proposed algorithm. Moreover, the released framework code includes agent training and evaluation to facilitate reproducibility and comparison.
>
> During the rebuttal period, we have further improved the work mainly in the following aspects:
> 1) **We conduct ablation studies on the key components of the algorithm to further clarify their contributions to performance.**
>
> As shown in **Figure 5 of Appendix A.6** (Ablation Experiment), we evaluate the components that drive performance in both the training and evaluation phases, including: MAPPO without Two-Stage, MAPPO without observation normalization, HATRPO, HAPPO, MAPPO, and MAPPO + Sequential Policy Updates. Based on the final evaluation results in **Table 3 of Appendix A.6** (Ablation Experiment), we conclude:
> - Observation normalization is crucial for stable con vergence; two-stage curriculum learning ensures the acquisition of post-pursuit behaviors such as energy management and exploration. Removing either component leads to task failure.
> - At 60 million steps, MAPPO suffers a training collapse caused by policy conflicts during training, which degrades performance. MAPPO + Sequential Policy Updates produces a smoother training curve and outperforms pure MAPPO in the final evaluation.
> - Building on Sequential Policy Updates, CSE-ET further incorporates an RNN, which improves performance and achieves the best results on both the training and evaluation curves.
> | Model | Rewards | TSR | PSR | ER |
> | :---: | :---: | :---: | :---: | :---: |
> | CSE-ET | $\\mathbf{1399.9049}_{\\pm 10.0}$ | $\\mathbf{0.7886}_{\\pm 1.1\\%}$ | $\\mathbf{0.6374}_{\\pm 1.8\\%}$ | $0.3440_{\\pm 2.2\\%}$ |
> | MAPPO | $1302.7120_{\\pm 14.0}$ | $0.7188_{\\pm 1.2\\%}$ | $0.5916_{\\pm 1.3\\%}$ | $\\mathbf{0.3703}_{\\pm 1.2\\%}$ |
> | HAPPO | $1094.2966_{\\pm 6.1}$ | $0.5874_{\\pm 1.1\\%}$ | $0.4506_{\\pm 1.2\\%}$ | $0.2450_{\\pm 1.2\\%}$ |
> | HATRPO | $1163.4772_{\\pm 3.4}$ | $0.6253_{\\pm 1.6\\%}$ | $0.5001_{\\pm 1.2\\%}$ | $0.2448_{\\pm 1.3\\%}$ |
> | Two Stage | $576.3578_{\\pm 17.7}$ | $0.0216_{\\pm 2.2\\%}$ | $0.0246_{\\pm 2.4\\%}$ | $0.0059_{\\pm 1.6\\%}$ |
> | Observation Normalization | $-15.7531_{\\pm 20.6}$ | $0.003_{\\pm 0.1\\%}$ | $0.003_{\\pm 0.15\\%}$ | $0.001_{\\pm 0.1\\%}$ |
>
> 2) **For independent critics, we add an experimental comparison by replacing the global critic in CSE-ET with independent critics:**
> | Model | **Rewards** | **TSR** | **PSR** | **ER** |
> | :---: | :---: | :---: | :---: | :---: |
> | CSE-ET | $\\textbf{1399.9049}_{\\pm 10.0}$ | $\\textbf{0.7886}_{\\pm 1.1\\%}$ | $\\textbf{0.6374}_{\\pm 1.8\\%}$ | $\\textbf{0.3440}\_{\\pm 2.2\\%}$ |
> | independent-critics | $956.2234_{\\pm 7.1}$ | $0.5245_{\\pm 1.1\\%}$ | $0.3872_{\\pm 1.2\\%}$ | $0.1621_{\\pm 1.2\\%}$ |
>
> 3) **We simulate noisy environments in the simulator by introducing the following two conditions:**
>
> - Increasing fluctuations in energy consumption to emulate a windy setting during testing, which reduces the reward of the energy-saving module.
> - Increasing fluctuations in the communication range to emulate a noisy channel, which reduces the reward of the exploration module.
> - The specific experimental results are as follows:
> | Model | **Rewards** | **TSR** | **PSR** | **ER** |
> | :---: | :---: | :---: | :---: | :---: |
> | CSE-ET | $\\textbf{1399.9049}_{\\pm 10.0}$ | $\\textbf{0.7886}_{\\pm 1.1\\%}$ | $\\textbf{0.6374}_{\\pm 1.8\\%}$ | $\\textbf{0.3440}_{\\pm 2.2\\%}$ |
> | w/ energy consumption | $906.7415_{\\pm 6.3}$ | $0.5216_{\\pm 1.2\\%}$ | $0.4046_{\\pm 1.3\\%}$ | $0.1831_{\\pm 1.1\\%}$ |
> | w/ noisy | $856.3421_{\\pm 9.5}$ | $0.5041_{\\pm 1.3\\%}$ | $0.3727_{\\pm 1.1\\%}$ | $0.1579_{\\pm 1.2\\%}$ |

---

> ### Author Response · Authors · 2025-11-24
>
> 4. **The paper has revised the following contents:**
> - We revised the description of $a\_{ij}$ and $\\beta$ in **Section 2.3**
> U2U Communication Model, so that the unit relations and connectivity conditions are clearer.
> - We added an explanation in **Section 2.4**
> UAV Power Consumption Model that Eq. (6) is a reasonable engineering approximation for modeling instantaneous power consumption, and we included experiments on $\kappa$ for validation, as shown in **Fig. 7**.
>
> We believe that the additional experiments, ablation studies, and improvements to the presentation further strengthen the contribution of this paper in terms of both novelty and practical applicability.
>
> We respectfully ask the reviewers to take these supplementary work and improvements into account in their final evaluation!

---

### Official Review · Reviewer_hCaU · 2025-11-02

**Soundness:** 3
**Presentation:** 3
**Contribution:** 2
**Rating:** 4
**Confidence:** 4

**Summary:**

This paper presents CSE-ET, a multi-agent reinforcement learning framework for cooperative 3D multi-UAV target tracking under constraints such as communication range, inter-UAV safety separation, and limited energy. CSE-ET integrates four key modules: a tracking model, an escape (evasion) model, a U2U (UAV-to-UAV) communication model, and a power consumption model. The main contribution is a three-layer system: a Base model, a Decision model that includes a derived 3D spatial information entropy, and finally, joint optimization performed using each agent having an actor network and a common shared critic, sequential policy updates, advantage decomposition, GAE, and PopArt. To explore the tradeoff between exploration and power consumption, the method integrates a tracking–exploration–power multi-objective reward and a curriculum-style fusion strategy. The simulation experiments conducted on NVIDIA Isaac Gym compare CSE-ET to MAPPO, HAPPO, and HATRPO under several reward weightings.

**Strengths:**

- The key strength of this work lies in the combination of UAV dynamics, evasion modeling, communication graph Laplacians, and detailed propulsion energy models.
- The 3-D extension and closed-form derivation of spatial information entropy are valuable.
- The two-stage curriculum learning and sequential policy update scheme describes how to handle multiple objectives within a single algorithm, enabling reward accumulation and faster convergence.
- The paper is clearly written and well-organized.

**Weaknesses:**

- Algorithmically, CSE-ET is a slight variant of MAPPO/HAPPO that incorporates sequential updates and normalization. Its novelty lies more in system integration (realistic two-phase reward design, energy and communication modeling, curriculum training) than in core learning methodology.
- The paper does not isolate the effect of sequential updates, reward shaping, or normalization. It remains unclear which component drives improvement. The paper would benefit from a controlled ablation study comparing pure MAPPO, MAPPO with sequential updates, MAPPO with PopArt, and full CSE-ET to isolate the effect of each modification.
- Other relevant baselines, such as graph neural networks, independent critics, or communication-aware MARL methods, have not been compared to.
- The experimental scope is narrow as it is limited to a single UAV tracking setup. Experiments are conducted in simulated open airspace, free from wind, sensor noise, communication delays, and obstacles.

**Questions:**

1. In what specific way does the CSE-ET algorithm diverge from MAPPO or HAPPO beyond sequential updates and normalization? Can an ablation study be performed? This would help clarify which component contributes to performance gains.
2. Are the values presented in Table 2 the mean or median of the experimental data? It could include the standard deviations to be complete.
3. How would the performance change when evaluated in noisy environments?
4. Has the algorithm been tested on a larger number of UAVs?

---

> ### Author Response · Authors · 2025-11-23
>
> # W1
> The goal of this work is to provide a training framework for multi-agent pursuit under strong partial observability and resource constraints that is closer to real-world needs, covers more scenarios, and is significantly more stable. To train agents that can pursue, explore, save energy, communicate, and collaborate, the stability of the training algorithm is crucial. Our key improvements and insights are as follows:
> - Independent actor networks (one per UAV) with sequential updates. MAPPO adopts shared-parameter training (a single actor network, all sampled data updated jointly). While this speeds up training, it can cause conflicts across multiple reward modules and lead to training collapse. As shown in Fig. 3.a, at 60 million steps, MAPPO with shared updates temporarily collapsed due to submodule conflicts in the multi-module reward. In CSE-ET, agents sample actions sequentially; the current agent’s local advantage conditions on the set of agents that have already acted (Eq. 25), yielding a global advantage function (Eq. 26). Over a long horizon of 100 million steps training agents with multiple complex cooperative behaviors, our algorithm shows superior stability compared with other baselines.
> - State-space normalization. The state space includes relative distance, relative velocity, pursuit angle, and escape angle. The raw feature scales differ greatly—for example, the relative-velocity range is [-50, 50], while angle ranges are [-2π, 2π]. Without normalization, large-scale features dominate the updates and small-scale features are ignored. This reduces algorithmic stability and can even prevent convergence.
>  - RNN-based feature extraction. By using hidden states to compress past information into a “smooth, low-dimensional state representation,” the TD error becomes more stable, improving overall algorithm stability. We also tried replacing the RNN with a transformer, but transformers require larger datasets and have many more parameters, making them more suitable for large-scale offline training and not well matched to our current on-policy setting.
> # Q1&W2
> We conduct ablation studies on the key components of the algorithm to further clarify their contributions to performance. PopArt normalizes the value-function targets across different tasks (pursuit, exploration, energy saving) and different phases (Phase I: accelerated pursuit; Phase II: steady pursuit; plus learning exploration and energy-saving behaviors). It is a widely validated, general-purpose component and is kept with a fixed configuration in all experiments below.
>
> As shown in **Figure 5 of Appendix A.6** (Ablation Experiment), we evaluate the components that drive performance in both the training and evaluation phases, including: MAPPO without Two-Stage, MAPPO without observation normalization, HATRPO, HAPPO, MAPPO, and MAPPO + Sequential Policy Updates. Based on the final evaluation results in **Table 3 of Appendix A.6** (Ablation Experiment), we conclude:
>
> 1) Observation normalization is crucial for stable convergence; two-stage curriculum learning ensures the acquisition of post-pursuit behaviors such as energy management and exploration. Removing either component leads to task failure.
>
> 2) At 60 million steps, MAPPO suffers a training collapse caused by policy conflicts during training, which degrades performance. MAPPO + Sequential Policy Updates produces a smoother training curve and outperforms pure MAPPO in the final evaluation.
>
> 3) Building on Sequential Policy Updates, CSE-ET further incorporates an RNN, which improves performance and achieves the best results on both the training and evaluation curves.
> | Model | Rewards | TSR | PSR | ER |
> | :---: | :---: | :---: | :---: | :---: |
> | CSE-ET | $\\mathbf{1399.9049}_{\\pm 10.0}$ | $\\mathbf{0.7886}_{\\pm 1.1\\%}$ | $\\mathbf{0.6374}_{\\pm 1.8\\%}$ | $0.3440_{\\pm 2.2\\%}$ |
> | MAPPO | $1302.7120_{\\pm 14.0}$ | $0.7188_{\\pm 1.2\\%}$ | $0.5916_{\\pm 1.3\\%}$ | $\\mathbf{0.3703}_{\\pm 1.2\\%}$ |
> | HAPPO | $1094.2966_{\\pm 6.1}$ | $0.5874_{\\pm 1.1\\%}$ | $0.4506_{\\pm 1.2\\%}$ | $0.2450_{\\pm 1.2\\%}$ |
> | HATRPO | $1163.4772_{\\pm 3.4}$ | $0.6253_{\\pm 1.6\\%}$ | $0.5001_{\\pm 1.2\\%}$ | $0.2448_{\\pm 1.3\\%}$ |
> | Two Stage | $576.3578_{\\pm 17.7}$ | $0.0216_{\\pm 2.2\\%}$ | $0.0246_{\\pm 2.4\\%}$ | $0.0059_{\\pm 1.6\\%}$ |
> | Observation Normalization | $-15.7531_{\\pm 20.6}$ | $0.003_{\\pm 0.1\\%}$ | $0.003_{\\pm 0.15\\%}$ | $0.001_{\\pm 0.1\\%}$ |
>
> **Table 3:** Final evaluation results of progressive techniques applied to CSE-ET.

---

> ### Author Response · Authors · 2025-11-23
>
> # Q2
> The data shown in Table 2 makes the average value, which has been modified to the result of adding standard deviation:
> | $\\boldsymbol{\\omega}\_{1}:\\boldsymbol{\\omega}\_{2}:\\boldsymbol{\\omega}\_{3}$ | **Rewards** | **TSR** | **PSR** | **ER** |
> | :---: | :---: | :---: | :---: | :---: |
> | 4:4:2 | $1399.6034_{\\pm 15.1}$ | $0.7979_{\\pm 1.1\\%}$ | $\\mathbf{0.7031}_{\\pm 1.5\\%}$ | $0.2984_{\\pm 1.1\\%}$ |
> | 4:2:4 | $917.5691_{\\pm 7.5}$ | $0.5379_{\\pm 2.2\\%}$ | $0.4367_{\\pm 1.1\\%}$ | $0.1941_{\\pm 1.2\\%}$ |
> | 4:3:3 | $1399.9049_{\\pm 10.0}$ | $0.7886_{\\pm 1.1\\%}$ | $0.6374_{\\pm 1.8\\%}$ | $\\mathbf{0.3440}_{\\pm 2.2\\%}$ |
> | 4:5:1 | $\\mathbf{1477.8414}_{\\pm 4.5}$ | $\\mathbf{0.8653}_{\\pm 1.3\\%}$ | $0.6581_{\\pm 1.2\\%}$ | $0.3307_{\\pm 2.1\\%}$ |
> | 4:1:5 | $56.8749_{\\pm 5.1}$ | $0.0334_{\\pm 6.2\\%}$ | $0.0239_{\\pm 4.1\\%}$ | $0.0118_{\\pm 5.0\\%}$ |
>
> **Table 2:** Agent (4:4:2) is recommended for missions prioritizing persistent tracking. Agent (4:3:3) is optimal for tasks requiring concurrent tracking and spatial exploration. Agent (4:5:1) should be deployed for emergency tracking scenarios.
> # W3
> We agree that graph neural networks (GNNs), independent critics, and MARL methods with explicit communication are relevant lines of work. We focus on HAPPO/MAPPO/HATRPO in the main manuscript for the following reasons:
>
> 1) Alignment between the problem setting and our contribution: Our contribution is a stable training framework for “strong partial observability + resource constraints” (with a unified objective and constraints over pursuit, exploration, energy saving, communication, and collaboration), rather than proposing a brand-new policy-gradient variant. HAPPO, MAPPO, and HATRPO belong to the on-policy trust-region family known for stability, which most directly tests whether our framework significantly improves robustness in training and deployment under the above constraints.
>
> 2) Fairness and reproducibility: These three methods share the same training paradigm—on-policy, centralized critic with decentralized execution (CTDE), and trust-region constraints—making ablations and controlled comparisons more comparable in terms of compute, sample efficiency, and convergence stability, and avoiding ambiguities in conclusions caused by cross-paradigm differences.
>
> For independent critics, we add an experimental comparison by replacing the global critic in CSE-ET with independent critics:
>
> | Model | **Rewards** | **TSR** | **PSR** | **ER** |
> | :---: | :---: | :---: | :---: | :---: |
> | CSE-ET | $\\textbf{1399.9049}_{\\pm 10.0}$ | $\\textbf{0.7886}_{\\pm 1.1\\%}$ | $\\textbf{0.6374}_{\\pm 1.8\\%}$ | $\\textbf{0.3440}\_{\\pm 2.2\\%}$ |
> | independent-critics | $956.2234_{\\pm 7.1}$ | $0.5245_{\\pm 1.1\\%}$ | $0.3872_{\\pm 1.2\\%}$ | $0.1621_{\\pm 1.2\\%}$ |
> # Q3&W4
> We simulate noisy environments in the simulator by introducing the following two conditions:
> 1) Increasing fluctuations in energy consumption to emulate a windy setting during testing, which reduces the reward of the energy-saving module.
> 2) Increasing fluctuations in the communication range to emulate a noisy channel, which reduces the reward of the exploration module.
> The specific experimental results are as follows:
> | Model | **Rewards** | **TSR** | **PSR** | **ER** |
> | :---: | :---: | :---: | :---: | :---: |
> | CSE-ET | $\\textbf{1399.9049}_{\\pm 10.0}$ | $\\textbf{0.7886}_{\\pm 1.1\\%}$ | $\\textbf{0.6374}_{\\pm 1.8\\%}$ | $\\textbf{0.3440}_{\\pm 2.2\\%}$ |
> | w/ energy consumption | $906.7415_{\\pm 6.3}$ | $0.5216_{\\pm 1.2\\%}$ | $0.4046_{\\pm 1.3\\%}$ | $0.1831_{\\pm 1.1\\%}$ |
> | w/ noisy | $856.3421_{\\pm 9.5}$ | $0.5041_{\\pm 1.3\\%}$ | $0.3727_{\\pm 1.1\\%}$ | $0.1579_{\\pm 1.2\\%}$ |
> # Q4
> We construct two- and three-UAV tracking setups in the simulation environment and conduct experimental validation; both exhibit collaborative tracking behaviors featuring exploration, energy saving, and communication. The experimental results are analyzed in Fig. 3, the simulation demonstration is shown in Fig. 1b, and video recordings are provided.

---

### Author Response · Authors · 2025-11-27

Dear ICLR 2026 **AC, SAC, and PC**

We would lilke to express our gratitude to all the reviewers for their valuable feedback. We have carefullyconsidered all suggestions and updated our submission accordingly.
However, we have not yet received responses from **Reviewer hCaU, Reviewer SGWU, and Reviewer uAAm** With only one week remaining for discussion, we kindly request your assistance in reaching out to thesereviewers. lt would be greatly appreciated if you could encourage them to review our rebuttal, as we areeager to know if we have adequately addressed their questions and concerns.
We believe that constructive and timely communication between, reviewers and authors is essential for thebenefit of both parties.
Thank you for your hard work and support.

Best regards,

The authors of Paper 5995

---

### Note · Authors · 2026-01-29

I have read and agree with the venue's withdrawal policy on behalf of myself and my co-authors.

---

### Meta-Review · Area_Chair_sgv1 · 2026-01-04

**Summary:**

This paper proposes CSE-ET, a multi-agent reinforcement learning framework for cooperative 3D UAV target tracking that aims to jointly optimize communication, spatial exploration, and power consumption under resource constraints. Despite addressing a complex multi-objective problem, the suggestion to **reject** is primarily informed by significant concerns regarding the soundness and novelty of the work. Reviewers critically noted that the underlying system models are unrealistic. Specifically, the kinematic formulation lacks the necessary degrees of freedom for accurate quadrotor control, and the power consumption model is incorrectly adapted from 2D flight literature to 3D maneuvers, severely undermining the validity of the simulation results. Additionally, the algorithmic contribution is viewed as a marginal variation of standard MAPPO/HAPPO architectures, with insufficient ablation studies to justify the performance gains or isolate the effects of the proposed sequential updates and normalization techniques. Given these fundamental issues with theoretical modeling and the limited experimental rigor, I recommend this paper for rejection.

**Reviewer Concerns:**

### Addressed Concerns
- Lack of Ablation Studies: The authors provided a new ablation study (Table 3, Appendix A.6) isolating the effects of sequential policy updates, observation normalization, and the curriculum learning strategy. This directly answered Reviewer hCaU’s request to identify which components drove performance.
- Insufficient Baselines: The authors addressed the request for comparison against "independent critics" by adding a specific experiment for it. They also justified why they did not compare against GNNs (a different paradigm).
- Narrow Experimental Scope: In response to claims that the environment was too sterile (no wind, single target), the authors added simulations with "noisy environments" (fluctuating energy/comms) and multi-UAV setups (2 and 3 agents).
- Dimensional Inconsistency (SNR vs. Distance): The authors successfully clarified that the "direct comparison" of SNR and distance (Reviewer uAAm's concern) was a notational shorthand for a threshold derivation. They provided the mathematical derivation to show the units are consistent in implementation.

### Unresolved Concerns
The critical concerns regarding the validity of the system modeling and algorithmic novelty (primarily from Reviewer uAAm) remain largely unresolved. The rebuttal clarified why the authors made certain choices, but confirmed that the models are indeed simplifications that arguably contradict the paper's claim of being "closely aligned with practical constraints."

- Invalid Power Consumption Modeling: One of the critical unresolved issues involves the validity of the power consumption model. Reviewer uAAm correctly identified that the authors apply a model derived from 2D forward flight literature (Zeng et al.) to a 3D rotary-wing scenario. This formulation depends solely on forward speed magnitude and completely neglects the substantial energy costs associated with vertical thrust and hovering—dominant factors in quadrotor dynamics. The authors' rebuttal, which characterizes this mismatch as a "reasonable engineering approximation" or "local linearization," fails to address the fundamental physical reality that 3D pursuit-evasion maneuvers cannot be accurately evaluated using a model that ignores vertical energy expenditure. **This flaw significantly undermines the paper's central claims regarding energy-efficient tracking.**
- Unrealistic Kinematics: The kinematic modeling of the UAVs remains insufficiently realistic for a study targeting robotics applications. Reviewers noted that the proposed model (Eq 1a-1c) treats the agents as point masses controlled by spherical acceleration vectors rather than true rigid-body quadrotors governed by 6-DOF dynamics (roll, pitch, yaw, and thrust). In their rebuttal, the authors acknowledged that their control variables merely parameterize acceleration direction rather than representing actual vehicle attitude. **This simplification confirms that the simulation does not capture the complex underactuated dynamics of real quadrotors**, directly contradicting the paper's assertion that the framework is "closely aligned with practical constraints."
- Arbitrary Evader Policy: The rigor of the evasion policy remains a significant point of contention. Reviewers criticized the evader model as heuristic and arbitrary, noting that it overlooks the extensive body of literature on differential games and optimal Hunter-Prey strategies. While the authors defended their approach as a "tunable" implementation combining trajectory smoothing and artificial potential fields, this defense confirms that the proposed framework is being tested against a sub-optimal, hand-crafted opponent. Relying on a heuristic evasion policy rather than a theoretically grounded optimal strategy weakens the evaluation of the tracking algorithm's robustness and performance limits.
- Incremental Novelty: The algorithmic contribution of the work is viewed as incrementally limited. The proposed CSE-ET framework is characterized by reviewers as largely a variation of standard MAPPO/HAPPO architectures, distinguished primarily by engineering adjustments such as sequential policy updates and observation normalization. While the authors' ablation studies demonstrate that these modifications improve training stability, they do not constitute a significant methodological innovation in multi-agent reinforcement learning. Without a strong theoretical advancement, the paper relies heavily on its "system integration" contribution. However, the value of this integration is heavily compromised by the aforementioned inaccuracies in the physical system modeling.

**Reviewer Scores:**

- Reviewer hCaU(4->4): While the reviewer would likely appreciate the new data and ablation studies, the core concern was that the method is a "slight variant of MAPPO" with limited novelty. The rebuttal improved the robustness of the experiments, but could not change the fact that the algorithmic contribution is incremental. So he/she may keep the original score for a rejection.

- Reviewer uAAm (2->2): The rebuttal would likely solidify this reviewer’s rejection. The authors admitted that their power model is a "local linearization" derived from 2D flight (ignoring vertical thrust energy cost). They admitted their kinematic model uses acceleration direction angles rather than true vehicle attitude. And they admitted the evasion policy is heuristic. These admissions serve as proof that the simulation does not accurately reflect the physical world. The "clarifications" effectively confirmed the reviewer's suspicion that the physics were oversimplified, making the results invalid for a robotics application paper.

- Reviewer SGWU (6->4): Reviewer SGWU’s initial positive assessment was heavily predicated on the belief that the paper offered a comprehensive solution to a "practical and complex" multi-objective problem. However, had SGWU engaged with Reviewer uAAm’s critique, he/she would have realized that the "practical" foundation of the paper is flawed. Reviewer uAAm demonstrated that the power consumption model (a core component of the multi-objective conflict) is invalid for 3D rotary-wing flight as it ignores vertical thrust. Consequently, the "complex conflicting environment" SGWU praised is actually utilizing a physically nonsensical reward function. Learning that the simulation does not accurately reflect the real-world physics would undermine SGWU’s confidence in the "soundness" of the results, leading them to downgrade the paper for failing to meet the practical standards they originally assumed were met.

---

### Decision · Program_Chairs · 2026-01-26

Reject